# High-resolution mass measurements of single budding yeast reveal linear growth segments

Andreas P. Cuny [1,2,8], K. Tanuj Sapra[1,7,8], David Martinez-Martin [1,3,4,8✉], Gotthold Fläschner [1,8], Jonathan D. Adams[1], Sascha Martin[5], Christoph Gerber[6], Fabian Rudolf[1,2✉] & Daniel J. Müller [1✉]

The regulation of cell growth has fundamental physiological, biotechnological and medical implications. However, methods that can continuously monitor individual cells at sufficient mass and time resolution hardly exist. Particularly, detecting the mass of individual microbial cells, which are much smaller than mammalian cells, remains challenging. Here, we modify a previously described cell balance ('picobalance') to monitor the proliferation of single cells of the budding yeast, *Saccharomyces cerevisiae*, under culture conditions in real time. Combined with optical microscopy to monitor the yeast morphology and cell cycle phase, the picobalance approaches a total mass resolution of 0.45 pg. Our results show that single budding yeast cells (S/G2/M phase) increase total mass in multiple linear segments sequentially, switching their growth rates. The growth rates weakly correlate with the cell mass of the growth segments, and the duration of each growth segment correlates negatively with cell mass. We envision that our technology will be useful for direct, accurate monitoring of the growth of single cells throughout their cycle.

[1] Eidgenössische Technische Hochschule (ETH) Zürich, Department of Biosystems Science and Engineering, 4058 Basel, Switzerland. [2] Swiss Institute of Bioinformatics (SIB), 4058 Basel, Switzerland. [3] The University of Sydney, School of Biomedical Engineering, NSW 2006 Sydney, Australia. [4] The University of Sydney Nano Institute (Sydney Nano), The University of Sydney, Sydney, Australia. [5] University of Basel, Department of Physics, 4056 Basel, Switzerland. [6] University of Basel, Swiss Nanoscience Institute (SNI), 4056 Basel, Switzerland. [7] Present address: Quantum-Si Inc, 530 Old Whitfield Street, Guilford, CT 06437, USA. [8] These authors contributed equally: Andreas P. Cuny, K. Tanuj Sapra, David Martinez-Martin, Gotthold Fläschner. ✉email: david.martinezmartin@sydney.edu.au; fabianrudolf@icloud.com; daniel.mueller@bsse.ethz.ch

Living cells sense and exchange biological, chemical, and mechanical information, as well as nutrients, water and waste products with their surrounding[1]. These processes, which are tightly linked to cell volume and mass, depend on cell state, growth, and division[2–5]. Cell volume and mass can be measured for larger cellular systems, such as adherent mammalian cells, tissues, and organs. However, a correlative method capable of continuously tracking at high resolution the total mass (<5%) of lighter cellular systems, such as single yeast cells whilst precisely monitoring their cell cycle state, has yet to be demonstrated[6–8]. Common methods used to approximate the volume of cells such as optical microscopy, Coulter counter devices, and flow cytometry, have linked cell size to cell type and state and correlate cell size with cell proliferation, gene expression, metabolism, disease, or death[2–4]. Considering that the regulation of cell volume and mass is physiologically essential, the dysregulation is connected to a broad range of diseases such as cancer, hypertrophies, or diabetes[4,6,9,10], and has consequences for biotechnological applications, including the cellular synthesis of biomolecular compounds or the control of cell growth and metabolism[11–13]. Thus, understanding of how cells regulate volume and mass is of substantial interest in the life sciences, systems biology, medicine, and biotechnology[4,14,15].

The growth of yeast has been studied for more than 60 years at the single cell level[16–18]. Despite many outstanding contributions, the fundamental question of how cell volume (or size) and mass is coupled to cell growth and division remains largely unanswered[9,15,19–23]. Particularly, whether growing yeast cells increase volume and/or mass linearly, exponentially, or otherwise is debated[19,20,24]. For example, *Saccharomyces cerevisiae*, a prominent model organism for cell size control studies, has been reported to increase volume non-linearly but its dry mass increases roughly linearly during cell generation[17]. Others have reported that both the total mass and volume increase exponentially[25]. Apparently, linear or exponential growth behaviors become more pronounced upon S-phase entry when *S. cerevisiae* initiates the replication and budding cycle[17,19,21,26]. Presumably, these conflicting findings originate from indirect measurements used to estimate cell size and dry mass[20] and/or from bulk measurements that simultaneously characterize many cells without taking their asynchronous cell cycle state into account[27]. It should be noted that the dry mass of a cell does not necessarily correlate to the cell volume (or size) nor to the total cell mass since water is a major contributor to the volume and total mass of a cell (≈60–80%)[8,27–30]. Moreover, assuming that the density of a cell is subject to relatively little fluctuations of <5%[15], it is tempting to calculate cell mass from cell volume measurements. However, even small variations in determining the radius (≈5–10%) of a spherical cell, which lie below the accuracy of common optical methods, lead to relatively large variations of cell volume and mass (≈15–30%). Such uncertainties in estimating the cell volume by optical methods increase even more for cells that adopt complex irregular shapes.

Over the last years, several promising nanotechnologies have been introduced to measure the mass of single cells, some of which attain resolutions below 1% of the cell mass[31]. Suspended microchannel resonators can measure the buoyant mass of floating cells with sub-femtogram resolution[32]. The buoyant mass of a cell is the total mass of the cell minus the mass of the fluid displaced by the cell. To monitor cell growth in full detail, it is advantageous to directly measure the total mass of a cell, which includes its water content. Suspended microchannel resonators can approximate the total mass of single cells by successively measuring their buoyant masses in fluids of different densities[33]. However, the approach assumes that the cell does not change in volume during the measurements, which each take ≈1–15 s, and

in response to fluid exchange[33]. Another challenge is to continuously measure the mass of single cells while they grow over hours or days. This can be achieved using pedestal mass sensors that can measure the total mass of adherent mammalian cells with an accuracy of ≈ 8 pg and a time resolution of ≈1 min[34]. Yet, their suitability for monitoring the mass of non-adherent and much smaller yeast cells has not been demonstrated.

Importantly, cell growth is tightly linked to cell morphology and state, and it is thus desirable to combine cell mass measurements with transmission light microscopy in order to simultaneously monitor cell mass, morphology and state. Recently we have established an inertial picobalance[7,35] which uses micromachined cantilevers[36–41] as mass sensors[42]. When combined with transmission light and fluorescence microscopy, the device can monitor the total mass and morphology of adherent mammalian cells. The measurements can be recorded over the time course of days, and reach ≈5 pg mass and 10 ms time resolution[7]. Thus far, however, the picobalance could not be applied to characterize much smaller, non-adherent yeast cells.

Here, we advance our inertial picobalance to simultaneously monitor the total mass and morphology of single yeast cells in culture conditions in real time. To be able to monitor single yeast cells progressing through the cell cycle, we increase the mass resolution of the picobalance by engineering small cantilevers as microresonators, introduce a new way to photoactuate the cantilevers with much lower laser power, and optimize the excitation and readout of the microresonators to maintain cellular proliferation under culture conditions. Further, we redesign the cantilever holder and fluid cell of the picobalance to enhance the signal-to-noise ratio of the combined fluorescence microscope so we can track the weak fluorescent signal of the yeast cell cycle proteins. The advanced picobalance, which now resembles a femtobalance, monitors the mass, morphology, and state of single yeast cells in real-time and at a resolution sufficient to study their growth and division at unprecedented detail. The unexpected insights into the growth behavior of single yeast cells guides to a refined growth model of yeast.

## Results

**Increasing mass resolution and fluorescent signal for monitoring single yeast cells**. The recently introduced picobalance achieved a mass resolution of ≈ 5 pg, which is sufficient to observe the growth of single adherent mammalian cells (≈ 2–4 ng)[7]. In order to characterize the growth of a single yeast cell, its total mass ranging from 10 pg for a newly budded cell to 100 pg for a mother cell[25], requires a higher mass resolution. To improve the mass resolution of the picobalance, we first considered the photoactuated microresonator, which resembles a cantilever beam (Fig. 1a). The natural resonance frequency of the cantilever $f_N = (2\pi)^{-1}\sqrt{k/m^*}$ depends on its spring constant $k$, and effective mass $m^*$, which comprises the mass of the cantilever and that of the yeast cell attached to the cantilever. The mass of a yeast cell can be quantified by tracking the difference of the natural resonance frequency ($\triangle f_N$) of the cantilever with and without the cell attached. The mass sensitivity of the picobalance, as given by the natural resonance frequency difference per unit of mass ($\triangle f_N/\triangle m$) depends on the effective mass (Fig. 1b). Therefore, lowering the mass of the cantilever by reducing its size increases the mass sensitivity of the measurement. However, reducing the cantilever size is limited because the cantilever must remain sufficiently large to host the cell and to separate the blue laser photoactuating the cantilever from the cell to prevent possible perturbations. Additionally, the intrinsic frequency noise of the measurement limits the minimum frequency difference $(\triangle f_N)_{min}$ and thus the minimum mass that can be detected. The noise

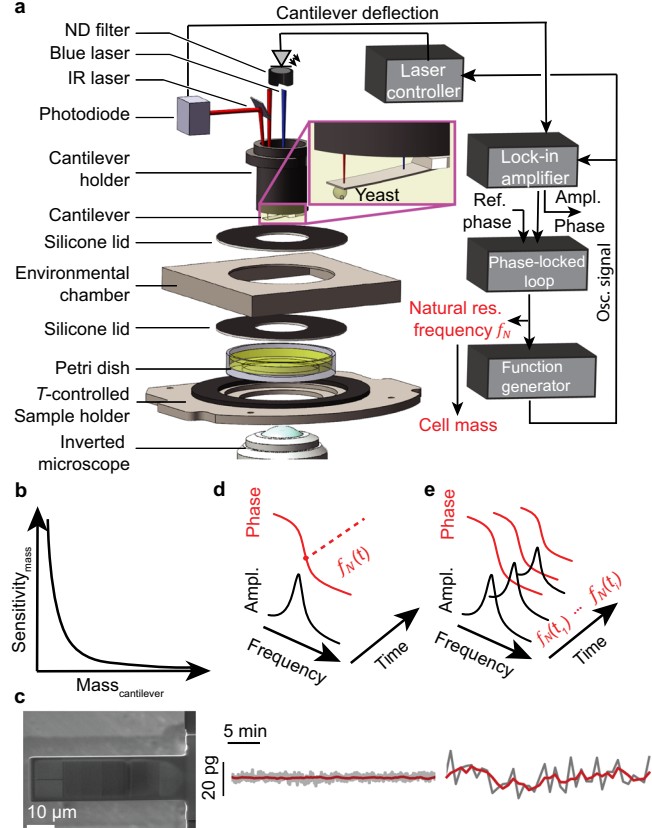

**Fig. 1 Experimental set-up for the simultaneous acquisition of mass and morphology of single yeast cells. a** Block diagram and main components of the picobalance. The total cell mass is detected by a cantilever that acts as microresonator and cell substrate. A blue laser photoactuates the cantilever at its natural resonance frequency $f_N$ and the cantilever movement is detected by an infrared (IR) laser reflected onto a photodiode. The blue laser power is reduced by a neutral density (ND) filter. Attaching a cell to the cantilever shifts $f_N$, which is tracked over time by keeping the cantilever phase at 90° by a lock-in amplifier, phase-locked loop, function generator and laser controller. A temperature (T)-controlled environmental chamber provides cell culture conditions (30 °C). The setup is placed on an inverted optical microscope as add-on module. Measurements are acquired in yeast culture medium. **b** The sensitivity of the cell mass detection is a function of the cantilever mass (see Eqs. (1) and (2), "Methods"). **c** Scanning electron microscopy image of a silicon nitride cantilever micromachined for yeast cell mass measurements. **d** Mass measurements ($n = 3$ from 3 independent experiments) using the continuous mode (only one measurement shown, all shown in Supplementary Fig. 1c). Top, to attain a high time and mass resolution, $f_N$ of the cantilever is continuously tracked by the phase-locked loop (see **a**). Bottom, a typical background measurement. On average, the noise is 2.3 ± 0.6 pg (mean ± SD) while smoothing (100 s moving window) reduces the noise to 0.45 ± 0.13 pg. **e** Mass measurements ($n = 3$ from three independent experiments) using the sweep mode (only one measurement shown, all shown in Supplementary Fig. 1d). Top, to minimize possible impact of the blue laser on cell viability, the cantilever is oscillated across frequency sweeps while recording the cantilever amplitude and phase. In between the frequency sweeps the blue laser is switched off. The sweep mode provides lower mass and time resolution. Bottom, a typical background measurement. On average, the noise is 11.0 ± 1.4 pg while smoothing (350 s moving window) reduces the noise to 4.6 ± 0.5 pg.

depends on the physical variables including the $f_N$, $k$, quality factor $Q$, oscillation amplitude $A$ of the cantilever, temperature $T$, and bandwidth (i.e., time resolution) $B$ of the measurement ("Methods").

To increase the mass resolution of the picobalance, we micromachined much narrower, shorter, and lighter silicon and silicon nitride microcantilevers by a focused ion beam than those previously used to measure the mass of adherent mammalian cells[7]. The new cantilevers were ≈ 60–70 μm long, ≈ 16 μm wide and ≈ 0.7–1 μm thick. The considerably reduced dimensions position a cell at the free end of the cantilever much closer to the blue laser photoactuating the cantilever base. To reduce possible perturbations of the yeast cell by the blue laser, we adapted our optical excitation scheme and photoactuated the cantilever by an intensity modulated ultra-low-powered blue laser (405 nm, ≈ 8–28 μW), whose power was ≈2–6 times lower than that used in our previous work[7] (Fig. 1a). To transfer minimal laser power to the cantilever and to prevent power instabilities of the laser diode, we operated the blue laser at high peak current and reduced the laser power by a non-reflective neutral density filter ("Methods"). To maximize the efficiency of photoactuating the cantilever with the considerably reduced blue laser power, we used piezo-motor-based nanopositioners to adjust the position and size of the blue laser spot at the base of the cantilever. Although the laser spot could approach ≈6 μm in diameter, we found that optimal photoactuation of the cantilever occurred at a diameter of ≈10 μm. With these measures, the ultra-low-powered and microspotted blue laser could photoactuate the cantilever to oscillate at its natural resonance frequency with amplitudes of ≈1–3 nm.

To monitor the natural resonance frequency of the cantilever, a low-powered infrared laser (852 nm, ≈165 μW) was reflected from the free end of the cantilever onto the photodiode. The amplitude and phase of the cantilever movement were analyzed using a lock-in amplifier (Fig. 1a). The cantilever phase, which measures the delay between the excitation signal and the mechanical response of the cantilever, is 90° at $f_N$. The phase was sampled every 10 ms by a phase-locked loop (PLL), which controlled a function generator to modulate the current driving the blue laser and thus the laser intensity oscillating the cantilever at $f_N$. We refer to this photoactuating mode as the continuous mode. The experimental set-up including the inverted optical microscope was placed in an acoustic isolation box maintained at 27.0 ± 0.1 °C. The picobalance was mounted on the microscope stage where an environmental chamber kept the cells at 30.0 ± 0.1 °C and prevented evaporation of the cell culture medium[7,43,44]. The experimental set-up showed an excellent long-term stability over the time course of hours without and with a cell being attached to the cantilever (Supplementary Fig. 1a, b), and approached a mass resolution of 2.3 ± 0.6 pg at a time resolution of 10 ms (Fig. 1d). Further, averaging the mass data over time windows of 100 s increased the mass resolution to ≤ 1 pg (0.45 ± 0.13 pg) thus approaching femtogram mass resolution (Supplementary Fig. 1c).

To correlate our mass measurements with the cell cycle phases more precisely, we wanted to fluorescently track cell cycle-specific proteins (Whi5 and Myo1)[45]. The combination of the relatively weak fluorescent signal and the relatively low numerical aperture of the objective (NA, 0.75) required to operate the cantilever at a working distance of ≈ 100–150 μm from the bottom of the Petri dish to prevent hydrodynamic effects, forced us to considerably enhance the signal-to-noise ratio of the fluorescence microscope of the picobalance. For this, we replaced the previously used polyether ether ketone (PEEK) cantilever holder with a black 3D printed acrylonitrile butadiene styrene (ABS) cantilever holder that featured a rough, non-reflective surface finishing. Additionally, we blackened the transparent silicone lids covering the environmental chamber and Petri dish (Fig. 1a). Finally, optimized filter sets for detecting the Myo1-mKate2 (3×) and Whi5-mKOκ (1×) enabled us to increase the signal-to-noise ratio of the weak fluorescent signal by a factor of ≈10 even with an objective of relatively low numerical aperture. Correlative mass

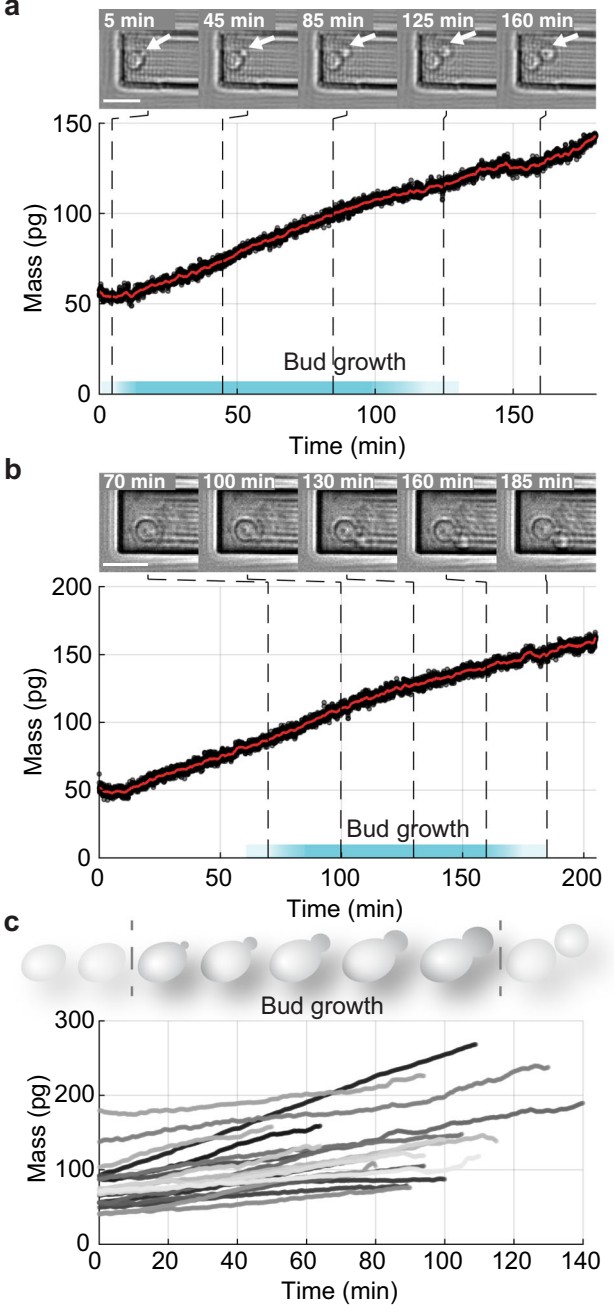

**Fig. 2 Mass and morphology of single *S. cerevisiae* cells budding daughter cells. a**, **b** While continuously measuring the total mass of a single yeast cell, differential interference contrast (DIC) images (taken every 5 min) show the budding of the cell attached to the cantilever of the picobalance (Supplementary Movie 1). White arrows indicate budding daughter cells. The raw data (black curve) shows the total mass of a growing yeast cell acquired every 10 ms using the continuous mode. The red curve shows the average raw data (100 s moving window, "Methods"). Cyan bars indicate where cells are in the S/G2/M phase when bud growth is observed. Yeast cells were attached to ConA-coated microcantilevers and the measurements were recorded in yeast culture medium at 30 °C ("Methods"). **c** Growth curves (*n* = 19) of single yeast cells in the S/G2/M phase (bud growth) as measured by the picobalance using the continuous mode (moving average of 100 s) in (*n* = 19) independent experiments. The overall growth rates between starting and end mass range between 0.2 and 1.6 pg min⁻¹, with an average of 0.6 ± 0.4 pg min⁻¹ (mean ± SD). The duration of the S/G2/M phase ranges from 50 to 140 min, with an average of 94 ± 23 min (Supplementary Fig. 4). DIC images in **a**, **b** were contrast enhanced using a custom flat field correction ("Methods"). Scale bars (white lines), 10 μm.

difference $\Delta f_N$ of the cantilever and accounting for the cell position[7,46]. To attach a single yeast cell, we functionalized the cantilever with the adherent substrate concanavalin A (ConA)[47,48]. After having measured $f_N$ of the ConA-functionalized cantilever, its free and tip-less end was vertically approached on a visually selected single yeast cell until a force of <1 nN was reached. The cantilever was kept in this position for ≈ 15 s to promote the attachment of the yeast cell and then retracted ≈ 100–150 μm from the bottom of the dish to avoid any hydrodynamic contribution of the dish during the mass measurements[49]. If needed, a purposely-made scratch on the dish supported this attachment by keeping the yeast in position (Methods). Starting from this time point the total mass and morphology of the cell were monitored simultaneously.

All mass measurements were performed using a prototrophic haploid *S. cerevisiae* strain cultured in pre-conditioned minimal synthetic defined medium with 2% D-glucose (SDmin) at 30.0 ± 0.1 °C ("Methods"). We picked up new-born yeast cells, and monitored their morphology (DIC imaging) and mass in real-time using the continuous mode (Fig. 2a, b and Supplementary Movies 1–3). The power of the blue laser photoactuating the cantilever oscillation at an amplitude of ≈ 1 nm was ≈ 8 μW. We recorded the total mass of single yeast cells at a time and mass resolution of 10 ms and ≈2.3 ± 0.6 pg, respectively. We then averaged the recorded mass data over a time window of 100 s, which increased the mass resolution to ≈ 0.45 ± 0.13 pg. At the beginning of the measurement, the total mass of the two exemplified yeast cells was ≈ 55 pg and ≈50 pg (Fig. 2a, b), whereas the mass of the 19 single yeast cells characterized in the G1/S transition ranged from ≈26 to 184 pg (Fig. 2c). Assuming that *S. cerevisiae* cells transiting from the G1 to the S/G2/M phase show diameters between 5 and 6 μm and densities of 1.1 g cm⁻³ [25,33,50], their total mass would range between 72 and 124 pg, which is in good agreement with our measurements (Supplementary Note 1).

After attachment to the cantilever the cells started to bud while increasing their mass with small non-uniform deviations from linearity (Fig. 2 and Supplementary Figs. 2 and 3). The measured average growth rate of the budding cells was ≈ 0.6 pg min⁻¹ (Fig. 2c). Estimating a growth rate from literature for the average cell cycle time[25,51], final cell volume[25,51] and density[25,50] for new-born *S. cerevisiae* cells leads to a growth rate of ≈0.4–0.6 pg min⁻¹, which is consistent with our measurements. However, close inspection of the individual mass curves recorded for each cell shows different growth rates (slopes) and times spent in the S/G2/

and fluorescence measurements were performed by recording frequency sweeps of the cantilever in 30 or 50 s intervals while switching off the lasers between the mass measurements for 20 s (Fig. 1e) and recording fluorescence images every 5 min. We introduced this mass measurement mode named "sweep mode" to minimize stress on the cell during fluorescent measurements and to corroborate mass measurements acquired using the continuous mode. The mass resolution using the sweep mode was 11.0 ± 1.4 pg and approached 4.6 ± 0.5 pg upon averaging the data with a sliding window of 350 s (Supplementary Fig. 1d). On average, the noise of mass measurements recorded using the sweep mode was roughly four times higher than in the continuous mode (Supplementary Fig. S1c, d).

**Monitoring mass and morphology of single yeast cells.** The picobalance determines the total mass of a yeast cell attached to a cantilever by measuring the natural resonance frequency

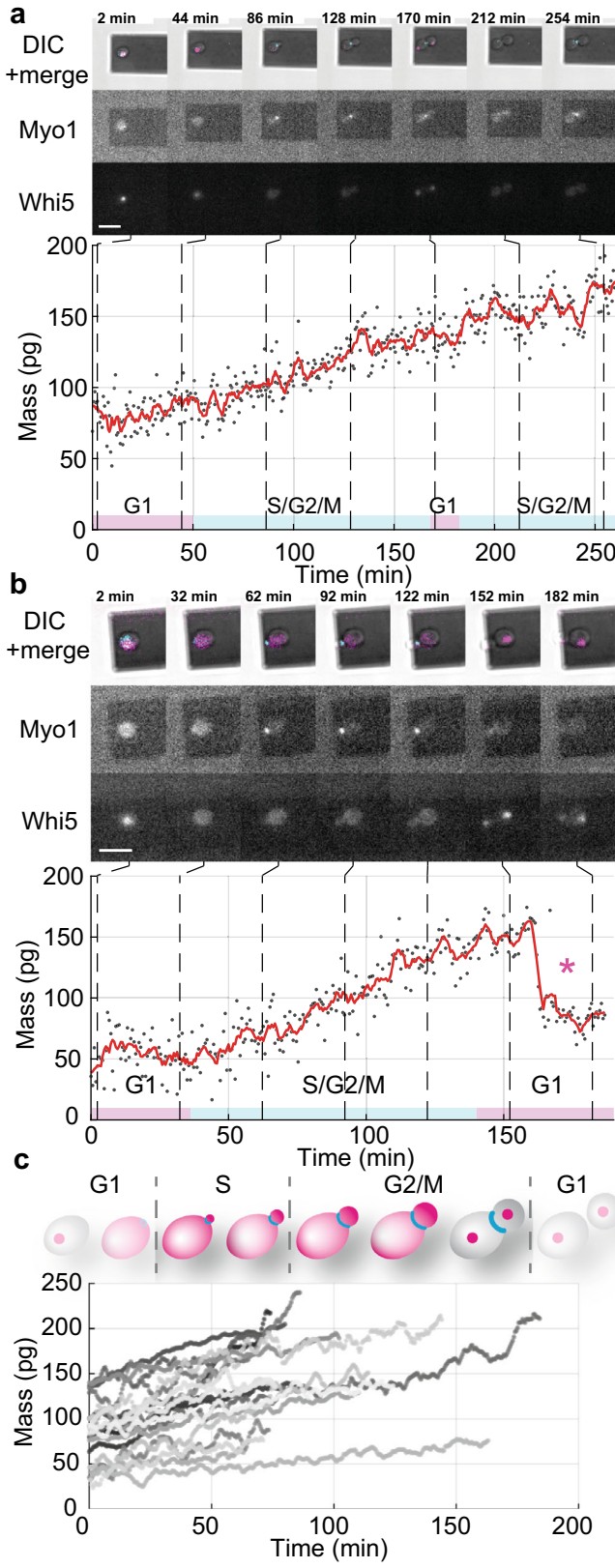

**Fig. 3 Mass and cell cycle measurements of single *S. cerevisiae* cells budding daughter cells. a, b** Single yeast cells expressing the fluorescently labeled cell cycle marker proteins (Myo1-mKate2 (3×) and Whi5-mKOκ (1×)), were imaged using differential interference contrast (DIC) and fluorescence microscopy every 2 min (upper panels). A phase and amplitude curve of the microcantilever were recorded over intervals ≈50 s to measure the cell mass using the sweep mode (Supplementary Movie 4). Between consecutive mass measurements, the infrared and blue lasers of the picobalance were switched off for ≈ 20 s to reduce bleaching of the fluorophores and to reduce potential perturbance of yeast growth. Cell mass values as derived from sets of single amplitude curves are shown as gray dots. Average raw data (350 s moving window, red line) shows the trend. Cyan bars on the time axis denote the S/G2/M phase of the yeast cell cycle, and magenta bars denote the G1 phase. The star (*) in **b** denotes the (partial) detachment of the daughter cell after cytokinesis, which drops the total mass. Scale bars (white), 10 μm. **c** Growth curves of (*n* = 19) single yeast cells progressing through the S/G2/M phase (bud growth) as measured by the picobalance using the sweep mode in (*n* = 19) independent experiments. The overall growth rates between starting and end mass range between 0.1 and 2.0 pg min⁻¹, with an average of 0.7 ± 0.5 pg min⁻¹ (mean ± SD). The duration of the S/G2/M phase ranges from 57 to 184 min, with an average of 96 ± 35 min.

relatively small cantilevers used in this work, we found that a laser power of ≈8 μW had no effect on the cell morphology and proliferation, even if the cell mass was monitored for several hours using the continuous mode (Fig. 2, Supplementary Figs. 2 and 3, and Movies 1–3). Taken together, the morphological changes, growth rate, and timing of the budding process are in good agreement with the literature for *S. cerevisiae*[17,25]. These results show that our high-resolution mass measurements maintain the viability of the yeast cells, as they grow and bud daughter cells.

**Monitoring mass and cell cycle phases of single yeast cells**. We next performed correlative mass measurements and fluorescence microscopy to better determine the cell cycle phases. Therefore, we fluorescently tagged the proteins Whi5 with mKOκ (1×) to determine the G1 phase and Myo1 with mKate2 (3×) to determine the S/G2/M phase[45] ("Methods"). Single cells from the yeast strain bearing the two cell cycle markers were imaged every 5 min by fluorescence microscopy and their mass monitored using the sweep mode (Fig. 3 and Supplementary Movie 4). At the beginning of the measurements the two exemplified single yeast cells showed a total mass of ≈75 pg (Fig. 3a) and ≈40 pg (Fig. 3b), whereas the mass of all 19 single yeast cells ranged from 33 to 136 pg (Fig. 3c). As observed above, each yeast cell progressing through the S/G2/M phase increased mass with small non-uniform deviations from linearity. The fluorescently tagged yeast cells progressing through the S/G2/M phase showed an average growth rate of 0.7 pg min⁻¹. The time to progress through S/G2/M was similar to the time measured using the continuous mode, that is, the time determined for non-tagged yeast cells (Supplementary Fig. 4). Moreover, the times needed for the yeast cells to progress through the S/G2/M phase are in agreement with reported values[17,51–56]. Occasionally, some mass measurements monitored yeast cells progressing through the relatively short G1 phase (Fig. 3a, b). However, the G1 phase was often not fully captured because of the too-long time needed (≈3–5 min) to optically select a yeast cell in the G1 phase and to attach the cell to the cantilever. Recurrently, the freshly budded daughter cell detached partially or fully from the mother cell after cytokinesis, which resulted in a mass drop (Fig. 3b, Supplementary Fig. 5, and Movie 5). We thus did not further analyze the growth of yeast

M phase, which highlights that individual yeast cells grow at different rates (Fig. 2c). This result is in agreement with previous findings that single yeast cells spend different times to progress through cell cycle phases[17,51–56]. On average, the cells spent ≈94 min in the S/G2/M phase (Supplementary Fig. 4). For the

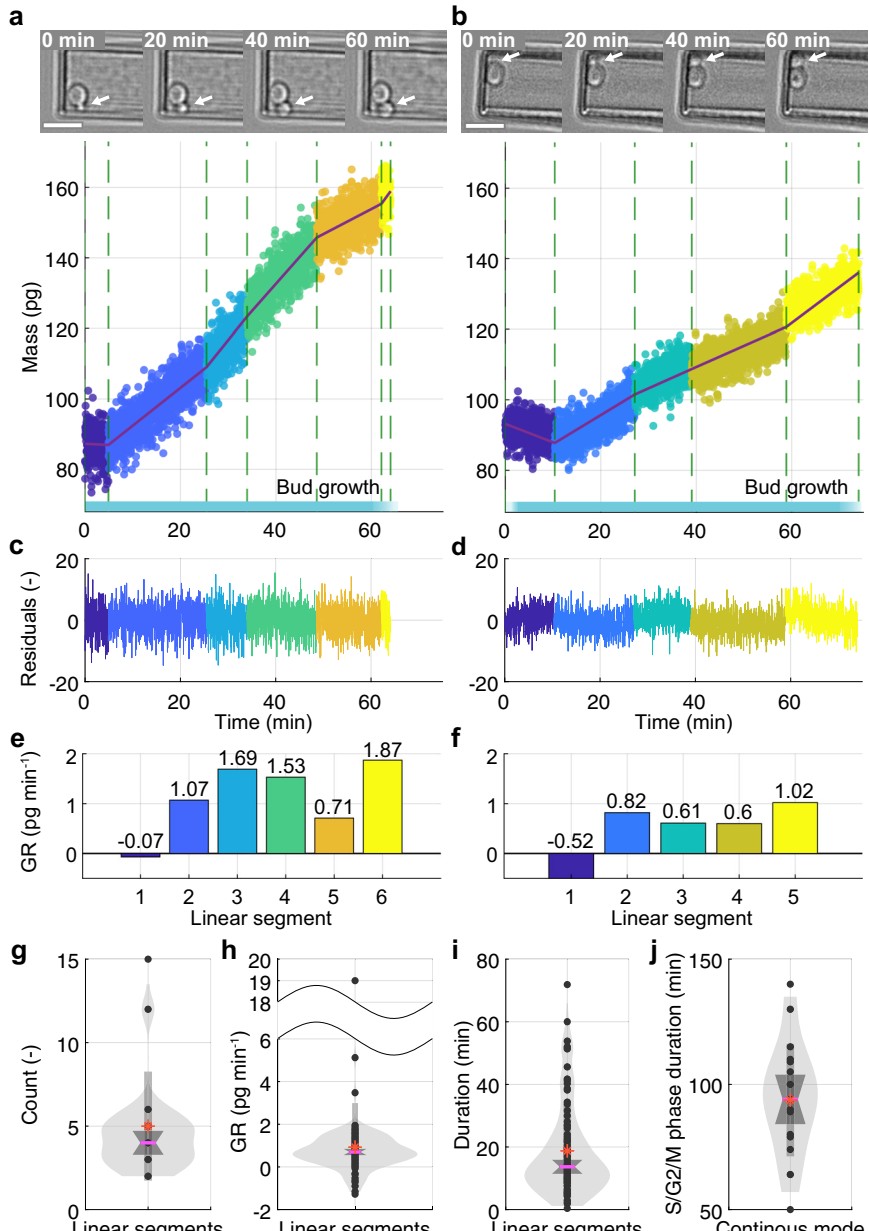

**Fig. 4 Single *S. cerevisiae* cells budding daughter cells increase mass in linear segments. a**, **b** Mass *versus* time curves of two *S. cerevisiae* cells segmented (green vertical dashed lines) into phases of linear growth (colored) by a segmented linear regression model (SLM) (violet lines). DIC images show budding cells attached to the cantilever of the picobalance. Shown are raw data. Scale bars (white), 10 μm. Mass measurements were recorded using the continuous mode. Cyan bars indicate where cells are in the S/G2/M phase when bud growth is observed. **c**, **d** Residuals of fitting the SLM to the raw data of growth curves. **e**, **f** Growth rate (GR) of each linear growth segment (colored) shown in (**a**, **b**). GR values given above the bars. **g**–**j** Growth analysis of ($n$ = 19) biologically independent cells measured using the continuous mode in ($n$ = 19) independent experiments. Data represented with violin plots show the raw data (gray dots), mean (orange star), median (pink line), mean and standard deviation (mean ± SD, medium gray vertical box), distribution as kernel density (light gray area from the 1st percentile $q$(0.01) to the 99th percentile $q$(0.99)) and the 95% confidence interval (CI, dark gray edges of hourglass hexagon). **g** Distribution of the number of linear growth segments measured for all cells. Mean is 5 (orange star) and median (horizontal pink line) is 4 segments per cell with 95% CI = [3.3, 4.7] (dark gray edges of hourglass hexagon). The distribution of the data is shown by the kernel density (light gray area) and the standard deviation (SD, medium gray vertical box). **h** Growth rates determined from linear growth segments of all cells. Median is 0.70 pg min$^{-1}$ with 95% CI = [0.6, 0.9]. **i** Duration (time span) of the linear growth segments of all cells. Median is 13.7 min with 95% CI = [11.4, 16.0]. **j** Duration of the budding (S/G2/M) phase of all cells. Median is 94 min with 95% CI = [84.1, 104.0].

cells progressing through the G1 phase (Supplementary Fig. 6). In summary, the control measurements performed using the sweep mode and in combination with fluorescence microscopy to better track the cell cycle phases confirm our cell mass measurements recorded using the high-resolution continuous mode.

**Single yeast cells show multiple segments of linear growth.** The growth curves recorded of single yeast cells using the continuous and the sweep mode (Figs. 2 and 3), show non-uniform deviations from linearity and thus from constant growth rates (Fig. 4). To investigate the growth curves in better detail, the high-

resolution growth curves (continuous mode) were fitted by either a segmented linear regression model (SLM), linear model (LM), or exponential model (EM) (Fig. 4 and Supplementary Fig. 7). We found that the SLM fits the observed growth of single yeast cells progressing through the S/G2/M phase best. The residuals of the SLM fit supports the goodness of the fit for the identified linear growth segments (Fig. 4a–d). Interestingly, the growth rate of each linear segment as determined by the SLM fit was seen to be higher or lower than that of its preceding or subsequent/neighboring segment (Fig. 4e, f and Supplementary Fig. 7). On average single yeast cells change mass in five distinct linear growth segments (Fig. 4g), each showing growth rates between 0.55 and 0.85 pg min$^{-1}$ (95% CI) and lasting on average for 18.7 min (Fig. 4h, i and Supplementary Fig. 7). We further validated our results by performing the same analysis of the mass curves of 19 yeast cells acquired using the sweep mode (Supplementary Fig. 8). Overall, almost every yeast cell showed multiple linear growth segments irrespective of whether their growth was monitored using the continuous or sweep mode with the few exceptions having been recorded at low mass resolution (Supplementary Figs. 7 and 9).

Our observation that single *S. cerevisiae* cells grow in consecutive segments of linear growth, with each segment having higher or lower growth rate than the preceding segment, provides important insight to the long-lasting debate of whether yeast cells grow linearly[17,57] or exponentially[18,25,58]. Although previous studies are difficult to compare due to different experimental and culturing conditions or insufficient resolution[18,59], our data show that at lower mass and time resolution, the details of the segmented linear growth behavior fade out (Supplementary Note 2). At the population level, we found that our cell growth data, after been normalized in mass and time for the S/G2/M phase, becomes compatible with a linear growth model (Supplementary Note 2). This effect could explain the rather controversial historical debate of whether yeast cells follow an overall linear or exponential growth behavior. Interestingly, previously published data on the volume increase of single yeast cells during the cell cycle could be described by several linear segments[26,45], which is compatible with the segmented linear increase of total mass of single yeast cells reported here. In summary, our results highlight (i) the need for high-resolution single cell technologies to study the growth behavior of single cells and (ii) the importance to align the growth data with cell cycle phases in order to avoid the individuality of the cell growth behavior to be lost.

**Correlations of single yeast cell growth and size.** Next, we investigated whether the growth rate of single yeast cells scales with cell size[27] and duration of the cell cycle phase[23,53,60] such as previously reported. Fitting the SLM model to our raw growth data approximated the growth rate and duration of every linear growth segment (Fig. 4 and Supplementary Figs. 7–9). At the single cell level, we found small correlations (Pearson correlation coefficient, $0.1 < r < 0.2$) of the growth rate of the linear segments with the cell mass after the G1/S phase transition (Supplementary Fig. 10a) or with the cell mass at cytokinesis (Supplementary Fig. 10b). We found a small-medium correlation ($r \approx 0.3$) of the growth rate of each linear segment with the cell at the start, average, or end time points of the segment (Supplementary Fig. S10c–e). We found a negligible correlation ($r = 0.08$) between the growth rate of linear segments with the duration of the S/G2/M phase and a small correlation ($r = 0.14$) and negative trend between the growth rate with the duration of each linear growth segment (Supplementary Fig. 10f, g). In addition, a negligible correlation was found between the duration of the S/G2/M phase and the cell mass at G1/S transition ($r = 0.08$) (Supplementary Fig. 10h). However, the strongest correlation, although being a medium correlation ($r = 0.37$), was a negative trend of the duration of the linear growth segments with the cell mass at the onset of the segment (Supplementary Fig. 10i). To summarize, the single cell growth rate increases minimally with the cell mass at the onset of the budding process and weakly depends on the cell mass at any point during the growth. Interestingly, however, the time a single cell spends in a specific linear growth segment depends inversely on its mass.

We also analyzed the growth rate of yeast cells at the population level similar to how it is routinely done upon analysing bulk measurements of cells[27,52,61]. We hence plotted the instantaneous growth rate *versus* cell mass of single yeast cells at different time points during the S/G2/M phase, and calculated their average growth rate, which showed a medium correlation to the data ($r = 0.4$) and strong correlation ($r = 0.9$) to the average of all single cells. (Supplementary Fig. 11a, b). Further binning the cell mass at the G1/S transition in six groups and averaging the growth rate during S/G2/M showed a strong correlation ($r = 0.6$) between cell mass and growth rate (Supplementary Fig. 11c). Thus, at the population level our averaged single cell growth data is consistent with literature[27] despite the clearly observed segmented linear growth at the single cell level. Further, no correlation between the yeast mass at the G1/S transition and the duration of the S/G2/M phase was observed at the population level (Supplementary Fig. 11d). To summarize, upon performing population level analyses, we find in agreement with literature[27] that on average cells with a higher mass (size) grow faster. However, at the single cell level we did not find strong correlation ($r > 0.5$) between cell mass and growth rate at G1/S transition. Overall, this finding suggests a more complex growth regulation at the single cell level, which probably would require experiments to be conducted at higher mass and time resolution and using more accurate cell cycle markers.

## Discussion

We have considerably improved our inertial picobalance to non-invasively record the total mass, morphology, and cell cycle of single yeast cells at high time and mass resolution. Key to achieving a higher mass resolution of $\leq 2.3$ pg at 10 ms and $\leq 0.5$ pg at 100 s time resolution, were to decrease the size of the cantilevers, and to increase the performance of the blue laser to photoactuate the cantilevers in the low power regime ($\approx 8\,\mu$W), such as needed for yeast to grow in an unperturbed manner. Overall, the mass resolution of the picobalance could be improved by a factor of $\approx 11$, which is $\approx 20$ times higher than the resolution achieved so far by pedestal mass sensors[34]. Additionally, we improved the signal-to-noise ratio of the combined fluorescence microscope by designing a cantilever holder that decreases straight and stray light in the optical path. These developments improved the performance of the picobalance together with advanced optical and fluorescence microscopy. By applying the continuous mode of the picobalance, an even higher mass resolution may be achieved by increasing the power of the blue photoactuating laser, which, however, can impair the viability of the yeast cell (Supplementary Note 3). Refining the mode of actuation and the geometry of the microresonator might further increase the mass resolution of the picobalance and suppress the influence of excitation and readout lasers on the sample viability. Operating the picobalance in a newly implemented sweep mode that switches off the excitation and readout laser of the picobalance between mass measurements minimizes light-induced stress on the light sensitive yeast cells and allows to record fluorescence

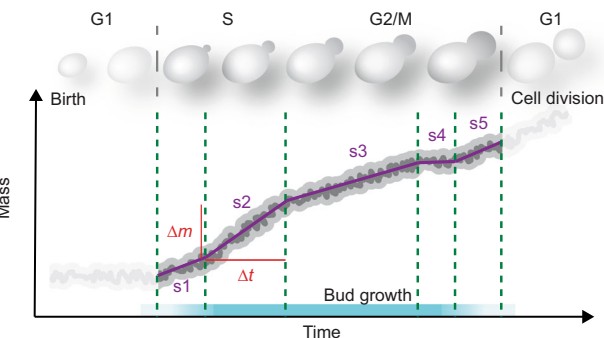

**Fig. 5. Model of a single yeast cell increasing mass throughout the S/G2/M phase in linear segments of constant growth rates.** The schematic of the budding yeast cell cycle above the plot ranges from the birth (G1 phase) to the budding phase (S/G2/M), the cell division, and the forthcoming G1 phase. During the S/G2/M phase, single *S. cerevisiae* cells increase mass $\Delta m$ in linear segments (purple lines), each lasting for a certain time $\Delta t$. Each linear segment (here s1–s5) shows a different growth rate (GR); $GR = \Delta m / \Delta t$ (Fig. 4) and lasts over a different time period $\Delta t$. At the resolution of our mass measurements, we observe single yeast cells to progress through the S/G2/M phase (cyan bar at bottom) on average in five linear growth segments.

signals at specific intervals (2–5 min in our case) such as needed to monitor the cell cycle state.

Despite many outstanding contributions for almost 100 years towards understanding cell growth regulation[62], fundamental aspects of how cell volume (or size) and mass evolve during cell growth and division remain elusive[4,19–21,63]. We believe that the main reasons for such conflicting results are indirect cell size or mass measurements made at insufficient time and/or mass resolution, together with over simplified assumptions, such as assuming a constant amount of water or dry mass over the life cycle of yeast cells[15,20]. Yet, these limitations arose naturally from the lack of methods to monitor the total mass of growing single cells directly and continuously at high resolution. Furthermore, where cell mass is measured, some ambiguity persists concerning the roles of dry, buoyant, or total mass and how these different masses relate to each other and to cell volume[20,31]. Here we used our microresonator-based picobalance to directly measure the total mass of single *S. cerevisiae*, which serves as a model organism in cell size control studies since more than 60 years[20]. By directly monitoring the total cell mass, morphology, and cell cycle phase, we observe that single *S. cerevisiae* cells progressing through the S/G2/M phase—in which they bud—grow in multiple linear segments (Fig. 5). Each linear growth segment shows a slope (growth rate) higher or lower than that of the previous segment and differ in timespan. For instance, some linear growth segments show a slope approaching to zero and hence no growth, while other segments show steep mass increases.

On average, single yeast cells progressing through the S/G2/M phase show five distinct linear growth segments. We identified a small-medium positive correlation between the growth rate of the linear growth segments with the average cell mass at each segment, and a medium negative correlation between the duration of the linear growth segments with the cell mass at the onset of the segment. These results suggest that yeast cells progress through the S/G2/M phase in a rather individualistic manner and that growth/mass regulation (and ultimately cell size control) is not exclusive for the G1 phase[64] but also in S/G2/M phase[53,65–67]. The observations have implications for bulk experiments, which characterize the growth of cellular ensembles. First, single cells need individual times to progress through the S/G2/M phase. Second, single cells show in average five linear growth segments,

which vary in an individualistic manner in growth rate and timespan. Consequently, bulk measurements cannot resolve these linear growth segments but rather observe a superimposition hereof. How this connects with the observed correlation between mother and daughter growth in size (volume)[22,27,66] remains to be explored. Bulk analysis of our mass data recorded during the S/G2/M phase, indeed shows a stronger positive correlation between the binned cell mass at the G1/S transition and the averaged growth rate during the S/G2/M phase as described[27]. Depending on the experimental parameters and methods used, this superimposition of the individualistic growth behaviors of single cells in bulk measurements can result in an apparently different growth behavior and is thus not suited to describe the rather individualistic growth behavior of single cells. Thus, to be able to observe the linear growth segments identified in this work, one must monitor the total mass of single yeast cells at sufficient time and mass resolution.

In summary, our results shed light into a long-lasting and lively-debated discourse of cell growth behavior and regulation[24]. It will be highly interesting to use the improved picobalance, to monitor how cells from different yeast species grow throughout their full and multiple cell cycles in different culture conditions. We believe that the presented method with its full integration with modern optical microscopy will enable the relationship of cell mass, size, and morphology to be studied in great detail. However, it may be speculated that upon further increasing the mass resolution of our picobalance and by improving the cell cycle markers one may be able to observe even more details of the linear growth segments by which yeast cells progress through their cell cycle. Finally, the fundamental question remaining to be solved is how do cells regulate the different linear segments of constant growth rates over their cell cycle and whether this regulation is inherited?

## Methods

**System setup.** The inertial picobalance comprises two customized lasers (Schäfter + Kirchhoff GmbH and Thorlabs) one blue laser having a wavelength of 405 nm being used to photoactuate the cantilever and one infrared laser with a wavelength of 852 nm being used to read out the cantilever movement. The blue laser brings along the advantage to efficiently photoactuate silicon (or silicon nitride) cantilevers even in the absence of metallic coatings, while the optical absorption coefficient of silicon decreases for longer wavelengths[68]. Because the wavelength of the blue laser is far away from the emission spectrum of common fluorophores, it enhances the compatibility of the picobalance with fluorescent microscopy. The power of the blue laser was set throughout the experiments to 8 μW and for the infrared laser to ≈165 μW. An exception was the high amplitude experiments where the power of the blue laser was set to 28 μW. The laser power was determined as average power using a photodiode power sensor S120VC (Thorlabs) connected to a power meter PM100D (Thorlabs). The intensity of the blue laser was modulated over time using a control current mode scheme implemented in a laser driver controller (LCD500, Stanford Research Systems). To avoid laser instabilities, the temperature of the blue laser diode was kept constant using a Peltier element, which was also driven by the laser driver controller. A neutral density filter with an optical density of 1.5 was used to reduce the optical power of the blue excitation laser and enabled to drive the laser diode well above the threshold in order to prevent phase instabilities. A four-quadrant Si PIN photo-diode (S5980 Hamamatsu) was used in combination with the infrared laser to read out the cantilever movement. To ensure that no other radiation could reach the photodiode, a hard-coated 25 nm bandpass optical filter (Edmund Optics) with a center wavelength of 850 nm was located just in front of the photodiode. A lock-in amplifier was used to extract the amplitude and phase of the cantilever movement. To track the natural resonance frequency of the cantilever at high temporal resolution (continuous mode) a PLL was integrated. Both the lock-in amplifier and PLL were parts of the picobalance prototype of Nanosurf AG (Switzerland). For the sweep mode, a custom-made LabVIEW 2019 (National Instruments) script was communicating with the laser driver of the blue laser via an RS232 connector whereas the LabVIEW script controlled the red laser directly using the Nanosurf scripting interface. The software operating the hardware and performing the measurements was the prototype software of the picobalance (version 2019, Nanosurf AG, Liestal, Switzerland).

A custom-made environmental chamber kept the sample at culture conditions and at 30.0 ± 0.1 °C during the experiments[7,43,44]. A customized and motorized stage (Nanosurf AG) accurately moved Petri dish and environmental chamber, so that individual cells could be easily picked up by the cantilever.

Picobalance, environmental chamber and motorized stage were mounted on an automated inverted microscope Nikon Eclipse Ti-E controlled by NIS Elements software (version 5.21, Nikon Europe B.V., Egg, Switzerland). The optical images of the DIC channel were taken with a Nikon 40× Plan Fluor objective, with a numerical aperture (NA) of 0.75 and a working distance of 0.66 mm, with module and prism for differential interference contrast. The illumination was based on a pE-100 LED diascopic light source (CoolLED Ltd., Andover, USA), filtered by a bandpass filter centered at 525 nm. The light was focused by a condenser with a NA of 0.29 and a working distance of 75 mm to accommodate our picobalance. The condenser was equipped with a polarizer/analyzer for differential interference contrast. For fluorescence microscopy, we equipped two filter cubes with the filters, excitation: 600/14 nm, beam splitter: 501 STHC 624 nm, emission: 655/40 nm for mKate2 and excitation: 546/6 nm beam splitter: ST565nm emission: 577/25 nm for mKOκ (AHF Analysetechnik, Tuebingen, Germany). The images were recorded by an ORCA Flash 4.0 V2 camera (Hamamatsu Photonic, Solothurn, Switzerland) with a pixel size of 6.5 microns, using 300 ms of exposure time per frame. 1×1 and 2×2 binning was used. The instrumental setup was enclosed by a sound and temperature isolating cabinet and the temperature within the cabinet was kept at $27.0 \pm 0.1\ °C$ to reduce the noise and the drift of the experimental setup.

**Mass resolution.** The mass sensitivity of the cantilever is given by:

$$\frac{\triangle f_N}{\triangle m} \sim \frac{1}{4\pi}\sqrt{\frac{k}{m^{*3}}} \qquad (1)$$

with the natural resonance frequency shift $\triangle f_N$ due to the attached mass $\triangle m$, the spring constant of the cantilever $k$ and the effective mass of the cantilever $m^*$[ref. [7]]. The minimal detectable frequency shift

$$\delta(\triangle f_N) = \sqrt{\frac{f_N k_B T B}{\pi k Q A^2}} \qquad (2)$$

with the natural resonance frequency $f_N$, Boltzmann constant $k_B$, the absolute temperature $T$, the temporal resolution (bandwidth, $B$), and the quality factor $Q$ and amplitude $A$ of the cantilever, is limited by thermal noise[69]. The minimal mass resolution is thus given by $\delta(\triangle f_N)$.

**Cantilevers.** Silicon (ARROW™ TL8, Nanoworld) and silicon nitride (HYDRA6R-200NG-TL, AppNano) cantilevers were micromachined using a focused ion beam to produce approximately 60–70 μm long, 16 μm wide, and approximately 0.7–1 μm thick cantilevers. The spring constant of the cantilevers was determined using Sader's method[70]. To attach yeast cells, cantilevers were functionalized with concanavalin A (ConA)[48]. Briefly, to prepare for functionalization, cantilevers were cleaned for 3 min in a bath of 95% sulfuric acid at room temperature. During cleaning, the cantilevers were gently moved manually to increase the efficiency of the cleaning process. Immediately after, the cantilevers were rinsed with ultrapure water (≈18 MΩ cm⁻¹) three times, and their chips blotted on precision wipes (Kimtech Science) for drying. The cantilevers were then placed for 15 min in an ultraviolet ozone cleaner (Jelight Company Inc.). Immediately after, the cantilevers were immersed for 1 h at 37 °C in a 250 μl droplet of 2 mg ml⁻¹ ConA (C2010 Sigma-Aldrich) in phosphate buffered saline (PBS). The cantilevers were then rinsed with PBS to remove weakly adsorbed ConA. Subsequently, the cantilevers were mounted in the picobalance and submerged in yeast medium. After functionalisation, the cantilevers were kept wet in between all steps.

**Cantilever holder.** To facilitate the epifluorescence microscopy of the weakly fluorescent yeast-strains, we used black acrylnitril-butadien-styrol (ABS) for the cantilever holder. ABS is biocompatible and contrary to polyether ether ketone (PEEK), which has been used previously[7], not auto fluorescent. We 3D-printed the holder with an Ultimaker S5 and 0.4 mm nozzle. The model was prepared with the software Cura (64 bit version 4.4.1.) with optimized extra fine 0.06 mm resolution. A support structure was printed with white breakout material in the second nozzle using standard settings. Brim was used as build plate adhesion type. Ironing was enabled to polish the top surface. We did not treat the printed cantilever holder with acetone or similar to keep a matt surface finish. We adapted the tolerances by reworking the cantilever holder manually to host the 10 mm diameter sapphire glass window (Edmund optics) and the screws to hold the cantilever clamp. The screws and the clamp of the holder were blackened with a camera varnish spray (TETENAL Europe GmbH, Art. Nr. 105202) with a light absorption of > 95% to reduce back reflected light.

**Data analysis.** All raw mass measurements have been analyzed with pyIMD[46] (version 0.1.3). For each mass measurement, a project file has been created containing the information related to a particular experiment such as cell position, spring constant of the cantilever etc. The computed total mass from the measured shift in the natural resonance frequency $f_N$ of the cantilever was saved in a comma separated (csv) file. For the sweep mode we modified pyIMD in such way that instead of fitting the recorded phases, the amplitudes were fitted, resulting in a lower noise. The amplitude was fitted according to $\frac{a}{\sqrt{(f^2 - f_N^2)^2 + \left(f \cdot \frac{f_N}{Q}\right)^2}} + b \cdot f + c$, with the frequency $f$, natural resonance frequency $f_N$, the driving amplitude $a$,

constants to compensate for a linear background $b$, c and the quality factor $Q$. The cell position on the cantilever was determined manually from the DIC images by measuring the distance from the free end of the cantilever to the center of mass of the cell. We used custom MATLAB (version R2021a, MathWorks, Natick, MA) scripts to further analyze and plot the data. To fit the segmented linear regression model (SLM) to the mass data we first determined potential initial rate change points with the function 'findchangepts' using linear statistics and a minimum threshold of $68 \times 10^3$ (continuous mode) and $2 \times 10^3$ (sweep mode). Finally, we used the function 'slmengine' with a degree of 1 and the previously obtained rate change seed points. We fitted linear models (LM) in the form of $y = \beta_0 + \beta_1 x$, exponential model (EM) as $y = \beta_0 \cdot e^{\beta_1 x}$. To enhance the contrast of the displayed microscopy images we used Fiji[71] (version 2.1.0). We first loaded all the images as sequence into a 16-bit stack. Then we cropped the image around the free end of the cantilever to focus on the dividing cell(s). We then duplicated the image stack and applied a Gaussian blur filter with $\sigma = 5$. We divided the raw image stack by the Gaussian blurred one and converted the resulting stack into 32-bit format. We then enhanced the contrast by setting the parameter saturated = 0.35 and the minimum and maximum to (0.37, 1.51). Finally, we ran an unsharp mask of radius = 1 and mask = 0.60 and converted the resulting stack to 8-bit. We saved the stack as a series of images, one per time point. For the statistical analysis of the S/G2/M phase durations we performed a two-sided t-test with the null hypothesis ($H_0$) that S/G2/M durations of continuous and sweep mode come from independent random samples from normal distributions with equal means and equal but unknown variance. Further, we performed a Kolmogorov–Smirnov test with null hypothesis ($H_0$) that the S/G2/M durations of continuous and sweep mode are from the same continuous distribution. For both tests, $H_0$ could not be rejected at the significance level alpha = 0.05. Effect sizes were reported as the Cohen's $d$ computed as:

$d = \frac{\bar{x}_1 - \bar{x}_2}{\sqrt{(s_1^2 + s_2^2)/2}}$, $s_i^2 = \frac{1}{(n-1)}\sum_{j=1}^{n}(x_{ij} - \bar{x}_i)^2$. Correlation coefficients were reported as Pearson correlation ($r$) computed as the square root of the $R^2$ of the respective linear fit.

**Yeast culture media.** Experiments were performed using the prototrophic haploid *Saccharomyces cerevisiae* strain FRY2540, which is a derivative of FY4of genotype MATa bearing a Myo1-mKate2 (3×) modification or strain FRY2795, which is a derivative of FY2540 of genotype MATa bearing a Myo1-mKate2 (3×) and Whi5-mKOκ (1×) modification[72–74]. For pre-culture, *S. cerevisiae* cells were grown in synthetic minimal defined medium (Smin) with D-glucose as sole carbon source (SDmin) at 30 °C in an orbital shaker. SDmin consists of 1.7 g l⁻¹ yeast nitrogen base from BD Biosciences (Germany), which does not contain amino acids or ammonium sulfate, 5 g l⁻¹ ammonium sulfate (Sigma-Aldrich Co., Germany), and 20 g l⁻¹ D-glucose (Sigma-Aldrich Co., Germany).

When cultivating yeast in the culture dish or on the cantilever of the picobalance, preconditioned media obtained by a single large batch was used to minimize the lag phase in single cell growth. To obtain this media, a 1.5 l culture was inoculated in a baffled 5 l Erlenmeyer flask at a density of $2 \times 10^5$ cells ml⁻¹. The cells were allowed to proliferate at 30 °C until reaching a density of $2 \times 10^6$ cells ml⁻¹. The media was then harvested by sterile filtration using a Steritop (Millipore) filter with a 0.22 μm pore size. The filtered media was stored in 50 ml aliquots at −20 °C.

**Yeast culture preparation.** Two days before every experiment, a 5 ml overnight pre-culture of SDmin medium was inoculated grown overnight on SDmin plates (SDmin + 20 g l⁻¹ agar, BD Biosciences, Germany). In following morning, the cells were diluted in fresh media to final concentration of 5–10 × 10⁶ cells ml⁻¹. The cell concentration was measured using a Z2 Coulter Counter (Beckman Coulter, Nyon, Switzerland) with an assumed doubling time of 1.2 h to calculate the dilution factor using custom software.

**Yeast culture in the picobalance.** For the first batch of experiments ($n_{cells} = 11$), a Petri dish (Ibidi) was gently scratched in different directions with sharp tweezers as described (Supplementary Fig. 12a). After that step, the dish was washed with dish soap, rinsed with ultrapure water and dried with nitrogen. In order to prevent the yeast from attaching to the bottom of the dish, the dish was filled with a solution of 2% bovine serum albumin (BSA, A3608 Sigma-Aldrich) in PBS for 30 min at 37 °C, to adsorb BSA on the bottom of the dish. For most experiments ($n_{cells} = 27$), the above described scratching has been skipped since with gaining experience the direct pickup of yeast cells from the untreated Petri dish became possible. To prepare the yeast cells for their attachment to the cantilever of the picobalance, the Petri dish was rinsed with PBS and filled with 1.5 ml of pre-conditioned medium at 30 °C. Finally, 20 μl of the yeast culture prepared as described above was diluted into the Petri dish. The Petri dish was then mounted in the controlled environmental chamber of the picobalance, where culture conditions were maintained at 30 °C.

**Yeast attachment to the cantilever.** The attachment of single yeast cells to the ConA-functionalized cantilever was performed 15–120 min after mounting the Petri dish with the cell culture, to give the yeast sufficient time to acclimatize in the picobalance. For this, the cantilever of the picobalance was approached onto a yeast

cell residing in the G1 phase as identified by optical microscopy (Supplementary Fig. 12b). Sometimes ($n_{cells} \approx 11$), the slightly tilted ($\approx 10°$) cantilever was brought into contact with a yeast cell and then moved laterally towards the edge of a scratch of the Petri dish in order to attach the yeast cell to the free end of the cantilever (Supplementary Fig. 12b–e). For most experiments ($n_{cells} \approx 27$), however, yeast cells were simply attached up by mechanically pushing the free end of the cantilever onto the cell (Supplementary Fig. 12c). To monitor mass, the cantilever with the adhering yeast cell was retracted from the bottom of the dish by $\approx$ 100–150 μm while still being constantly immersed in yeast culture medium at cell culture conditions (Supplementary Fig. 12f).

**Yeast cell cycle determination**. The different phases of the cell cycle were determined by tagging the proteins Myo1 and Whi5 with fluorescent proteins (-mKate2 (3×), -mKOκ (1×) respectively). Myo1 is present during bud neck formation, starting in late G1 phase and used as a proxy for start and end of the S/G2/M phase. Whi5 relocalizes into the nucleus during late M phase until end of G1 and is used as a proxy for the G1 phase[45]. During this study, the presence (or absence) of the signal from the two markers was quantified manually to determine the cell cycle phase for each dataset. Movies of mass curves including the extracted cell cycle information were generated with a custom MATLAB (version R2021a, MathWorks, Natick, MA) script. The image contrast enhanced of the raw images of the fluorescent channel as well as the creation of the false colored channel merge (DIC, mKate2, mKOκ channel) we used Fiji[71] (version 2.1.0). We loaded each channel as sequence into a 16-bit stack. We then used "Merge Channels…" to create a false colored composite image. We assigned the DIC channel to gray, the mKate2 to cyan, and mKOκ to magenta. Then we cropped the composite image stack around the free end of the cantilever to focus on the dividing cell(s). The contrast was adjusted for each channel to enhance the visibility of the signal and to suppress the background fluorescence. We converted the resulting composite to 8-bit and saved the stack as a series of images, one per time point, channel, and color composite.

**Reporting summary**. Further information on research design is available in the Nature Research Reporting Summary linked to this article.

## Data availability

All data generated and used in this study in its raw and processed form have been deposited in the ETH Research Collection (https://doi.org/10.3929/ethz-b-000547242).

## Code availability

The scripts required to reproduce all data representations and statistics are included in the data repository organized in a per figure/movie basis for computational reproducibility (ETH Research Collection, https://doi.org/10.3929/ethz-b-000547242). The software used to analyze the raw data (pyIMD, version 0.1.3) has been previously described in detail[46] and is available for download from Gitlab (https://gitlab.com/csb.ethz/pyIMD/tree/master).

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

## Acknowledgements

We thank D. Mathys for assisting with scanning electron microscopy and focused ion beam lithography; A. Azizoğlu for support with the yeast cultures; A. Tonin and the electronic workshop of the physics department (University of Basel) for helping with the beam deflection electronics; the mechanical workshop of the physics department (University of Basel) for their help with the development and manufacturing of the device; P. Buchmann and P. Argast from the electronic and mechanical workshop of the department of biosystems science and engineering (ETH Zurich) for helping with the temperature controlled system of the cabinet; L. Gonzalez and G. König for the controller upgrade, bug fixing, and experimental consultation; G. König for supporting long-term stability tests; D. Skibinski for basic LabView implementations; H.-M. Kaltenbach and L. Dharmarajan for discussions regarding data analysis; R. Newton for critically reading and discussing the initial manuscript; Nanosurf AG for their technical support customizing mechanical and electronic components. This work was supported by the NCCR Molecular Systems Engineering, the Innosuisse (project 28033.1 PFNM-NM), and the Swiss Nanoscience Institute Basel.

## Author contributions

D.M.-M., K.T.S., A.P.C., G.F., F.R., and D.J.M. designed the experiments. The picobalance prototype used in this work was adapted and built using the original picobalance as template by D.M.-M., S.M., and J.D.A. with help from C.G. and D.J.M. The picobalance prototype was modified as detailed in the following. D.M.-M., J.D.A. and S.M. modified and optimized the driving laser system. A.P.C., G.F., and K.T.S. designed, engineered, and applied the new cantilever holder to conduct mass measurements with improved fluorescence microscopy. D.M.-M. designed the high mass resolution cantilevers. K.T.S. finetuned the cantilevers for optimized resonance. J.D.A. adapted the lock-in amplifier and programmed the control software. K.T.S. and D.M.-M. conducted continuous and sweep mode mass measurements with optical microscopy. K.T.S. conducted mass measurements (sweep mode) with fluorescence microscopy. Mass measurements were conducted with the help of G.F. and A.P.C. A.P.C. engineered the yeast strains. A.P.C. cultured the yeast strains with D.M.-M., K.T.S., and G.F. A.P.C. and G.F. wrote the data analysis pipeline. D.M.-M. suggested the SLM analysis, which was implemented and performed by A.P.C. A.P.C. and G.F. performed the population level analysis and G.F. simulated the linear and exponential population growth models. A.P.C. wrote the pipeline to plot the data and scripts to enhance the contrast of the microscopy images. A.P.C., D.M.-M., K.T.S., G.F., D.J.M., and initially F.R. discussed and iterated the experiments, data analysis, and figures. A.P.C., D.M.-M., K.T.S., G.F., D.J.M., and initially F.R. wrote the manuscript. All authors read and approved the manuscript.

## Competing interests

D.M.-M., S.M., C.G., and D.J.M. have filed two patents related to the technology of the cell picobalance and its applications (US20170052211A1 and WO/2015/120,991). D.M.-M., G.F., S.M., and D.J.M. have filed a patent related to the environmental chamber (US10545169B2). G.F. and J.D.A. both joined Nanosurf AG, which commercializes the picobalance. The remaining authors declare no competing interests.
