## [Peer Review File · Nature Communications]

Reviewers' Comments:

Reviewer #1:

Remarks to the Author:

The paper by Martinez-Martin and colleagues report on measuring accurately the growth of yeast cells and on observing non-monotonous growth patterns. The subject of cell growth is of broad interest and advancing methods for accurate cell mass measurements are welcome. The authors report on a resonating balance which works on adherent cells. I have two comments that in my opinion are important, as follows.

1. The authors should discuss the effects of dissipation due to the vibration taking place in a solution. Showing some raw power spectrum data would be informative, as the line broadening due to loss would be visible.

2. The method seems to be almost identical with the one presented in a previous report by the authors:

Martinez-Martin, D. et al. Inertial picobalance reveals fast mass fluctuations in mammalian cells. *Nature* 550, 500-+, (2017).

Are there points of novelty that can be emphasized, other than shaving mass off the cantilever?

Reviewer #2:

Remarks to the Author:

Here, the authors present the adaptation of a pico-balance to monitor the mass and therefore growth parameters of budding yeast. This issue of how cells accumulate mass is a long standing and important question. Despite some controversy it has been generally understood that different yeast species accumulate mass or volume in some approximately exponential manner. Here, the authors report the development of technology and methodology for adapting a previously reported pico-balance to measure single yeast cells. They then go on to propose that budding yeast mass accumulation is not exponential but is linear with a pause in growth prior to budding. This is a potentially interesting result, but I have three (related) reservations about the authors conclusion regarding yeast mass accumulation that need to be addressed before publication.

First, I think only a very small number of cells are being measured. As far as I can tell the authors are drawing conclusions regarding the growth patterns in budding yeast from a single cell (!). Needless to say it is impossible to advocate publication of the authors conclusions without a robust number of cells. To generate any remotely meaningful assessment of the patterns of yeast growth I think ~20 cells would be required, whereas even more would be ideal. Getting this number of cells is presumably straightforward for the authors given their statement in the abstract: "Considering the ease of use our technology we envision that it will be widely applied . . ."

Second, mass accumulation of larger cells was previously reported to be faster than for smaller cells (eg Godin et al., 2010 *Nat Methods*). This is generally assumed to be because cells grow closer to exponentially than to linearly (thereby explaining why a culture of small mutant cells can double in number with close to the same generation time as a culture of larger mutant cells). If, as the authors propose, cells do not grow exponentially it requires that larger cells have a higher linear growth rate than smaller cells, which the authors need to test as a minimum requirement to clarify and corroborate their conclusion.

Third, as the authors state when measuring at high resolution after "60 min the budding process stopped (Supplementary Fig. 3 and Movie 4), indicating that the increased laser power impaired cell growth over an extended period of time". This does call in to question whether the growth in the proceeding 60 minutes is likely to be representative of a truly wild-type growth patterns. The authors need to address this potential issue with their methodology in some way (less laser power?).

Reviewer #3:

Remarks to the Author:

Two years ago Martinez-Martin and co-workers introduced the inertial picobalance (8) to weigh and detect mass fluctuations on adherent cells. That paper was a breakthrough in nanomechanical sensing. The inertial picobalance enables to follow in real time the mass changes of a living cell. In that paper, the authors took great care in designing a very sensitive device compatible with a single cell biology.

The current manuscript is a follow up. The mass sensitivity of the inertial picobalance has been improved by reducing the total mass of the cantilever. This was achieved by reducing the cantilever size. The current inertial picobalance is more reliable, robust and sensitive than the first prototype. This evolution might be necessary for commercial or pre-commercial applications (refs. 31-33). However, it seems that all the necessary physics and/or technology involved in reducing the cantilever size has been borrowed from other fields (AFM).

The technology associated with the present contribution is outstanding. Moreover, the authors findings are explained with clarity in the manuscript. Nonetheless, I find it hard to determine how this paper will influence thinking in the field. This manuscript relies heavily of their previous results. The fact that the mass sensitivity has been improved to the picogram range is an achievement. However, it is far from surprising. Nanomechanical resonators fabricated by top-down approaches have achieved mass sensitivities in the zeptogram range in vacuum (Yang, Y. T. et al. Nano Lett. 6, 583–586 (2006)). Closer to the manuscript context, ref. 28 claims a mass resolution in the sub-femtogram range. In my view, the novelty of the manuscript rests on the relevance of the observed stepwise growth of a budding yeast cell. The authors should provide more details about this point.

Comments.

1 The field of nanomechanical sensors has 20 years of history. Cantilever-based sensors have been applied to measure biological interactions and mass changes (DNA, proteins, cells). There are some good reviews (for example, J. Tamayo et al. Chem. Soc. Rev. (2014)) and original contributions. Some of those references could strengthen the context of the manuscript.

2 Unprecedented accuracy is repeated several times. It appears twice in the abstract.

3 Photothermal excitation can be accomplished with a range of light visible wavelengths (blue, red...). If there is a technical/scientific reason for using blue light, this should be mentioned.

4 In page 12 is stated that ‘...this stepwise linear growth...’. The terms ‘stepwise’ and ‘linear’ are at odds. Stepwise means it proceeds in steps (finite jumps). Linear means proportional to a parameter (here time).

5 Similarly, the meaning of void slope is obscure. Do the authors mean a negative slope (Fig. 3) ?. Flat slope ?.

6 Figure 3, could the authors the place an arrow at the times that there is a step in the growth ?.

7 How many yeast cells show a stepwise mass growth?. Is Fig. 3 representative of the growth cycle of budding yeast cells ?. I might be wrong, but Figure 4 does not seem to correlate with the pattern shown in Fig. 3 (time: 15 to 45 minutes).

Reviewer #4:

Remarks to the Author:

In this manuscript the authors describe the development of a molecular ‘balance’, capable of continuously measuring the mass of single yeast cells at pg resolution. They use the information they gathered from monitoring in real time dividing cells to arrive at the conclusion that yeast cells increase in mass linearly, not exponentially as it is often assumed. If true, these results have important implications for current models of size control and how cell growth is coupled to division, because a linear increase in mass (documented with the most accurate instrument possible) questions the need/existence for critical size controls during cell division. Overall, the authors appear to have significantly ‘moved the needle’ in the field regarding the instrumentation they developed to measure the mass of small cells. This is the main reason for being excited about the paper and its potential impact. However, I have several concerns:

1. It was not clear that the authors measured the increase in mass in the G1 phase of the cell cycle. It appears that for single cells their measurements begun at the time of bud emergence (before that and during attachment to the cantilever the cells did not grow in their setup). If true,

their measurements essentially exclude the G1 phase. This is crucial, because much of the interest in the linear vs. exponential model of growth is in G1.

2. Their value of 92 pg/cell at the beginning of the experiment appears too large. Certainly much larger than the value of ~20 pg in the paper they cite by Mitchison's measurements of dry mass. I guess adding the missing liquid will bring that value closer to ~60 pg, but still their 92 pg value appears too high.

3. The authors need to check more strains and media to further validate their results. For example, A haploid vs. diploid strain in rich vs. poor media will provide important reference points.

4. The authors do not mention some key references:

- For the significance of linear vs. exponential growth in terms of cell size control, perhaps one of the most lucid papers is the now classic Conlon and Raff paper (PMID: 12733998). A plethora of newer papers, especially as they relate to the 'adder' phenomenon should also be discussed.
- For linear growth (measured in dry mass) they cite the 1958 Mitchison paper, perhaps the 1964 paper from Scopes and Williamson (PMID14195443) should also be cited.
- Linear growth (in size), was also observed by the Heinemann group with a microfluidic device during the entire lifespan of yeast cells (PMID22421136).
- On the other hand, the Manalis group showed that growth (in size), especially in G1, is exponential (PMID22272256).

Point-by-Point Response to the Reviewer's Comments

Reviewer #1

Reviewer #1: The paper by Martinez-Martin and colleagues report on measuring accurately the growth of yeast cells and on observing non-monotonous growth patterns. The subject of cell growth is of broad interest and advancing methods for accurate cell mass measurements are welcome. The authors report on a resonating balance which works on adherent cells. I have two comments that in my opinion are important, as follows.

Authors: Thank you for your encouraging and constructive comments. It was indeed not easy to develop the theory and instrumentation to monitor the growth of single yeast cells with a microcantilever at high mass sensitivity and time resolution. As kindly acknowledged by the reviewer we also expect the technological approach and biological insight presented to be of broad interest for life scientists, biotechnologists and medicines focussing on the understanding of cell growth. Below we explain point-by-point how we addressed each concern of the reviewer in our revised Manuscript.

Reviewer #1: 1. The authors should discuss the effects of dissipation due to the vibration taking place in a solution. Showing some raw power spectrum data would be informative, as the line broadening due to loss would be visible.

Authors: Thank you. The oscillating (vibrating) cantilever is exposed to higher dissipation in solution compared to in air. The higher dissipation reduces the quality factor Q of the cantilever resonance from ≈ 115 in air to ≈ 3 in cell culture medium (\approx aqueous solution). The lower the quality factor the wider the amplitude resonance curve and the flatter the phase resonance curve. The cantilever phase is not defined when the cantilever is not directly excited but just subjected to thermal noise. Below we show the dissipation associated to the quality factor Q of the cantilever by showing the cantilever amplitude and phase curves in air and cell culture medium (SDmin) (**Fig. R1**). This considerable reduction of Q in aqueous solution tends to reduce the mass sensitivity of the oscillating cantilever by limiting the minimum frequency shift $\delta(\Delta f_N)$ of the cantilever that can be detected according to the following expression:

$$\delta(\Delta f_N) = \sqrt{\frac{f_N k_B T B}{\pi k Q A^2}} \quad \text{Eq. R1}$$

with k_B being the Boltzmann constant, T the absolute temperature and B the measurement bandwidth (i.e., temporal resolution), and f_N being the natural resonance frequency, k the spring constant and A the oscillation amplitude of the cantilever¹. To achieve a high mass resolution requires to detect small changes of the frequency shift Δf_N . Guided by Eq. R1, we enhance the mass resolution by increasing the cantilever

oscillation amplitude A with the intensity modulated blue laser². Exciting the cantilever enables the cantilever's phase to be defined. We refer to this discussion, which we had published earlier (see ref³ including supplementary information), in our revised manuscript and more clearly describe our technological approach to (see revised Manuscript, **Results** "Increasing mass resolution and fluorescent signal for monitoring single yeast cells" and **Methods**, "Mass resolution").

Since the power spectrum of the cantilever (thermal noise) has an undefined phase, which is the signal we require to for high resolution mass measurements, we show here both the amplitude and phase response of the cantilever in air and cell culture medium (**Fig. R1**). The response show the increased dissipation in cell culture medium (lower quality factor ≈ 3 in liquid *versus* ≈ 115 in air). As explained above, the detrimental effect of higher dissipation in the mass resolution is compensated by increasing the oscillation amplitude of the cantilever (above the thermal noise).

Fig. R1. Cantilever amplitude and phase response in air and in cell culture synthetic defined minimal medium (SDmin). Measured raw amplitude and phase data as a function of the frequency (grey crosses) and the fit (red line). The estimated quality factor Q in air is 114.91 for the amplitude or 115.11 for the phase. In cell culture medium (minimal synthetic defined media (SDmin)), the Q factor is 3.37 for the amplitude or 3.32 for the phase, respectively.

Reviewer #1: 2. The method seems to be almost identical with the one presented in a previous report by the authors:
Martinez-Martin, D. et al. Inertial picobalance reveals fast mass fluctuations in

mammalian cells. Nature 550, 500-+, (2017). Are there points of novelty that can be emphasized, other than shaving mass off the cantilever?

Authors: The reviewer comments that the operational method of our picobalance is almost identical to that presented in our previous report, which we had applied to measure the inertial (total) mass of adherent mammalian cells³. We kindly apologize if we did not describe at sufficient clarity the technological inventions that were necessary to be able to non-invasively monitor the inertial mass of much smaller and non-adherent yeast cells. To address this challenge, we had to introduce the following key technological novelties: *(i)* We had to modify the optical design of the picobalance in order to avoid that the blue laser photoactuating the much smaller cantilever perturbs yeast cell growth. To reach this goal and to stably oscillate of the cantilever at much lower (blue) laser power the blue laser had to be reduced ≈ 6 times ($\approx 8\mu\text{W}$) compared to our previous work for non-invasive conditions. *(ii)* In order to determine the cell cycle phases more precisely, we had to combine fluorescence microscopy of yeast cells with mass measurements (**revised Fig. 1, new Fig. 3, new Supplementary Movie S4**). To achieve this goal, we developed a new cantilever holder to enhance the signal-to-noise ratio of the fluorescence microscope. Further, to combine the single yeast cell fluorescence measurements with mass measurements we had install a mass measurements mode (sweep mode) in order to allow the yeast cells to growth in an unperturbed manner (see **revised Results** “*Increasing mass resolution and fluorescent signal for monitoring single yeast cells*” and “*Monitoring mass and cell cycle phases of single yeast cells*” and **revised Methods**). *(iii)* We had to design cantilevers having approximately half of the mass than in our previous work while keeping their natural resonance frequency in aqueous solution at $\approx 50 - 70$ kHz. The newly designed much smaller cantilevers allowed us to observe single yeast cells in cell culture medium at ≤ 2.3 pg mass and ≈ 10 ms time resolution. *(iv)* We had to further develop a data processing pipeline, which allows to reduce the mass resolution to ≤ 0.45 pg at a time resolution of 100 s. Despite of using a much reduced laser power to photoactuate the cantilever, the mass resolution of the picobalance could thus be increased by a factor of ≈ 11 . To quantitatively compare the technological advancement made with literature, the (total) mass resolution of our technologically improved picobalance is now ≈ 20 times better than the (total) mass resolution achieved by pedestal mass sensors⁴.

Revised Fig. 1 of the revised Manuscript. Experimental set-up for the simultaneous acquisition of mass and morphology of single yeast cells. **a**, Block diagram and main components of the picobalance. The total cell mass is detected by a cantilever that acts as microresonator and cell substrate. A blue laser photoactuates the cantilever at its natural resonance frequency f_N and the cantilever movement is detected by an infrared (IR) laser reflected onto a photodiode. The blue laser power is reduced by a neutral density (ND) filter. Attaching a cell to the cantilever shifts f_N , which is tracked over time by keeping the cantilever phase at 90° by a lock-in amplifier, phase-locked loop, function generator and laser controller. A temperature (T)-controlled environmental chamber provides cell culture conditions (30°C). The setup is placed on an inverted optical microscope as add on module. Measurements are acquired in yeast culture medium. **b**, The sensitivity of the cell mass detection is a function of the cantilever mass (see Eq. 1 and 2, **Methods**). **c**, Scanning electron microscopy image of a silicon nitride cantilever micromachined for yeast cell mass measurements. **d**, Mass measurements using the continuous mode. Top, to attain a high time and mass resolution, f_N of the cantilever is continuously tracked by the phase-locked loop (see **a**). Bottom, a typical background measurement without a cell. On average, the noise is 2.3 ± 0.8 pg (mean \pm s.d.) while smoothing (100 s moving window) reduces the noise to 0.45 ± 0.16 pg. **e**, Mass measurements using the sweep mode. Top, to minimize possible impact of the blue laser on cell viability, the cantilever is oscillated across frequency sweeps while recording the cantilever amplitude and phase. In between the frequency sweeps the blue laser is switched off. The sweep mode provides lower mass and time resolution. Bottom, a typical background measurement without a cell. On average, the noise is 11.0 ± 1.7 pg while smoothing (350 s moving window) reduces the noise to 4.6 ± 0.6 pg.

New Fig. 3 of the revised Manuscript. Mass and cell cycle measurements of single *S. cerevisiae* cells budding daughter cells. **a, b,** Single yeast cells expressing the fluorescent cell cycle markers (Myo1-mKate(3x) and Whi5-mKOκappa(1x)), were imaged using DIC and fluorescence microscopy every 2 min (upper panels). A phase and amplitude curve of the microcantilever were recorded over intervals ≈ 50 s to measure the cell mass using the sweep mode (**Supplementary Movie S4**). Between consecutive mass measurements, the infrared and blue lasers of the picobalance were switched off for ≈ 20 s to reduce bleaching of the fluorophores and to reduce potential perturbation of yeast growth. Cell mass values as derived from sets of single amplitude curves are shown as grey dots. Average raw data (350 s moving window, red line) shows the trend. Cyan bars on the time axis denote the S/G2/M phase of the yeast cell cycle and magenta bars denote the G1 phase. The star (*) in **b** denotes the (partial) detachment of the daughter cell after cytokinesis, which drops the total mass. **c,** Growth curves ($n = 19$) of single yeast cells progressing through the S/G2/M phase (bud growth) as measured by the picobalance using the sweep mode. Scale bars, 10 μm .

Taken together, we believe that these technological improvements provide a considerable advancement to the field. The high resolution both in mass and in time at reduced laser power allowed us to obtain quite unexpected insight that single yeast cells progress through the S/G2/M phase via distinct segments of constant growth rate. We have revised our manuscript to clearly outline these technological advancements and the new biological insights revealed (see **revised Abstract, Introduction, Discussion and Methods of the Manuscript**). For example the **revised Results** now reads:

*“To increase the mass resolution of the picobalance, we micromachined much narrower, shorter and lighter silicon and silicon nitride microcantilevers by a focused ion beam than those previously used to measure the mass of adherent mammalian cells³. The new cantilevers were $\approx 60\text{--}70\ \mu\text{m}$ long, $\approx 16\ \mu\text{m}$ wide and $\approx 0.7\text{--}1\ \mu\text{m}$ thick. The considerably reduced dimensions position a cell at the free end of the cantilever much closer to the blue laser photoactuating the cantilever base. To reduce possible perturbations of the yeast cell by the blue laser, we adapted our optical excitation scheme and photoactuated the cantilever by an intensity modulated ultra-low-powered blue laser (405 nm, $\approx 8\text{--}28\ \mu\text{W}$), whose power was $\approx 2\text{--}6$ times lower than that used in our previous work³ (**Fig. 1a**). To transfer minimal laser power to the cantilever and to prevent power instabilities of the laser diode, we operated the blue laser at high peak current and reduced the laser power by a non-reflective neutral density filter (**Methods**).”*

and

*“We refer to this photoactuating mode as the continuous mode. The experimental set-up including the inverted optical microscope was placed in an acoustic isolation box maintained at $27.0 \pm 0.1\ ^\circ\text{C}$. The picobalance was mounted on the microscope stage where an environmental chamber kept the cells at $30.0 \pm 0.1\ ^\circ\text{C}$ and prevented evaporation of the cell culture medium^{3,5,6}. The experimental set-up showed an excellent long-term stability over the time course of hours with and without a cell being attached to the cantilever (**Supplementary Fig. S1a,b**), and approached a mass resolution of $2.3 \pm 0.8\ \text{pg}$ at a time resolution of 10 ms (**Fig. 1d**). Further, averaging the mass data over time windows of 100 s increased the mass resolution to $\leq 1\ \text{pg}$ ($0.45 \pm 0.16\ \text{pg}$) thus approaching femtogram mass resolution (**Supplementary Fig. S1c**).”*

and

*“To correlate our mass measurements with the cell cycle phases precisely, we wanted to fluorescently track cell cycle specific proteins (Whi5 and Myo1)⁷. The combination of the relatively weak fluorescent signal and the relatively low numerical aperture of the objective (NA, 0.75) required to operate the cantilever at a working distance of $\approx 100\text{--}150\ \mu\text{m}$ from the bottom of the Petri dish to prevent hydrodynamic effects, forced us to considerably enhance the signal-to-noise ratio of the fluorescence microscope of the picobalance. For this, we replaced the previously used polyether ether ketone (PEEK) cantilever holder with a black 3D printed acrylonitrile butadiene styrene (ABS) cantilever holder that featured a rough, non-reflective surface finishing. Additionally, we blackened the transparent silicone lids covering the environmental chamber and Petri dish (**Fig. 1a**). Finally, optimized filter sets for detecting the Myo1-mKate(3x) and Whi5-mKOk enabled us to increase the signal-to-noise ratio of the weak fluorescent signal by a factor of ≈ 10 even with an objective of relatively low numerical aperture (NA, 0.75). Correlative mass and fluorescence measurements were performed by recording frequency sweeps of the cantilever in 30 or 50 s intervals while switching off the lasers between the mass measurements for 20 s (**Fig. 1e**) and recording fluorescence images every 5 min. We introduced this mass measurement mode named “sweep mode” to*

minimise stress on the cell during fluorescent measurements and to corroborate mass measurements acquired using the continuous mode. The mass resolution using the sweep mode was 11.0 ± 1.7 pg and approached 4.6 ± 0.6 pg upon averaging the data with a sliding window of 350 s (Supplementary Fig. S1c). On average, the noise of mass measurements recorded using the sweep mode was roughly four times higher than in the continuous mode (Supplementary Fig. S1c,d)."

Point-by-Point Response to the Reviewer's Comments

Reviewer #2

Reviewer #2: Here, the authors present the adaptation of a pico-balance to monitor the mass and therefore growth parameters of budding yeast. This issue of how cells accumulate mass is a long standing and important question. Despite some controversy it has been generally understood that different yeast species accumulate mass or volume in some approximately exponential manner. Here, the authors report the development of technology and methodology for adapting a previously reported pico-balance to measure single yeast cells. They then go on to propose that budding yeast mass accumulation is not exponential but is linear with a pause in growth prior to budding. This is a potentially interesting result, but I have three (related) reservations about the authors conclusion regarding yeast mass accumulation that need to be addressed before publication.

Authors: Thank you for your encouraging and constructive comments, which guided us to revise our Manuscript. Below please find our point-by-point response to the specific referee comments.

Reviewer #2 1: First, I think only a very small number of cells are being measured. As far as I can tell the authors are drawing conclusions regarding the growth patterns in budding yeast from a single cell (!). Needless to say it is impossible to advocate publication of the authors conclusions without a robust number of cells. To generate any remotely meaningful assessment of the patterns of yeast growth I think ~20 cells would be required, whereas even more would be ideal. Getting this number of cells is presumably straightforward for the authors given their statement in the abstract: "Considering the ease of use our technology we envision that it will be widely applied."

Authors: Thank you. Being encouraged by the reviewer and despite facing severe challenges due to the pandemics, we have considerably improved the methodology and analysis, and have increased the number of independent experiments to 38 yeast cells. In addition, the necessary controls to confirm the experimental results were increased substantially (**new Fig. 2, new Fig. 3, new Supplementary Fig. S1–S5**). The new data, which has been included in the revised Manuscript, supports and extends our previous results described in our initially submitted Manuscript. Most excitingly the unexpected biological insights advance the current growth model of yeast cells (see **new Fig. 4, new Supplementary Fig. S7–S9**). In general, we observe that single yeast cells show individualized growth behavior, which is best described by multiple segments of linear growth (**new Fig. 4**). Each such linear growth segment is defined by a constant growth rate (mass change per time), which can be either higher or lower than the growth rate of the adjacent segments. The segmented linear growth model, is now outlined and discussed in our **revised Manuscript** (see **revised Abstract, Results, Discussion** and **new Fig. 5**).

Encouraged by the reviewer's comment and the new results achieved upon characterizing the growth of a higher number of yeast cells, we measured the mass of yeast cells, which had been fluorescently tagged with Myo1 to monitor the S/G2/M phase and with Whi5 to monitor the G1 phase of the cell cycle (**new Fig. 3**). The mass measurements acquired using the high-resolution continuous mode (**new Fig. 2**) in the absence of fluorescence microscopy is confirmed by the mass measurements recorded using the independent sweep mode in the presence of fluorescence microscopy (**new Fig. 3**, and **Supplementary Fig. S5**). We find that the growth of yeast cells is not impaired by either one of our mass measurement modes (**Supplementary Fig. S4**). Correlation of the data recorded in the continuous mode and sweep mode further shows that we can determine the S/G2/M phase of single yeast cells also in absence of the fluorescent cell cycle markers at sufficient accuracy (**new Supplementary Fig. S4**). Our extended experiments and the data analysis of the mass measurements of single yeast cells advance previous growth models, which were mainly derived from analysing yeast cells at the population level (**new Supplementary Fig. S14, Supplementary Note 2**). Our experimental data shows that single cells increase mass in several linear segments each having a different constant growth rate (**new Fig. 4, new Supplementary Fig. S7–S9**). Our results, which are summarized in an advanced growth model (**new Fig. 5**), also suggests that previous, partially misleading cell growth models have been concluded from experiments that do not provide sufficient mass and time resolution of single yeast cells.

New Fig. 2 of the revised Manuscript. Mass and morphology of single *S. cerevisiae* cells budding daughter cells. a, b, While continuously measuring the total mass of a single yeast cell, differential interference contrast (DIC) images (taken every 5 min) show the budding of the cell attached to the cantilever of the picobalance (**Supplementary Movie S1**). White arrows indicate budding daughter cells. The raw data (black curve) shows the total mass of a growing yeast cell acquired every 10 ms using the continuous mode. The red curve shows the average raw data (100 s moving window, **Methods**). Cyan bars indicate where cells are in the S/G2/M phase when bud growth is observed. Yeast cells were attached to ConA-coated microcantilevers and the measurements were recorded in yeast culture medium at 30°C (**Methods**). **c,** Growth curves ($n = 19$) of single yeast cells in the S/G2/M phase (bud growth) as measured by the picobalance using the continuous mode (moving average of 100 s). The overall growth rates range between 0.2 pg min^{-1} and 1.6 pg min^{-1} , with an average of $0.7 \pm 0.3 \text{ pg min}^{-1}$ (mean \pm s.d.). The duration of the S/G2/M phase ranges from 50 min to 140 min, with an average of $93 \pm 21 \text{ min}$ (**Supplementary Fig. S2**). DIC images in **a** and **b** were contrast enhanced using a custom flat field correction (**Methods**). Scale bars, 10 μm .

New Fig. 3 of the revised Manuscript. Mass and cell cycle measurements of single *S. cerevisiae* cells budding daughter cells. a, b, Single yeast cells expressing the fluorescent cell cycle markers (Myo1-mKate(3x) and Whi5-mKOkappa(1x)), were imaged using DIC and fluorescence microscopy every 2 min (upper panels). A phase and amplitude curve of the microcantilever were recorded over intervals ≈ 50 s to measure the cell mass using the sweep mode (**Supplementary Movie S4**). Between consecutive mass measurements, the infrared and blue lasers of the picobalance were switched off for ≈ 20 s to reduce bleaching of the fluorophores and to reduce potential perturbation of yeast growth. Cell mass values as derived from sets of single amplitude curves are shown as grey dots. Average raw data (350 s moving window, red line) shows the trend. Cyan bars on the time axis denote the S/G2/M phase of the yeast cell cycle and magenta bars denote the G1 phase. The star (*) in **b** denotes the (partial) detachment of the daughter cell after cytokinesis, which drops the total mass. **c**, Growth curves ($n = 19$) of single yeast cells progressing through the S/G2/M phase (bud growth) as measured by the picobalance using the sweep mode. Scale bars, 10 μm .

New Fig. 4 of the revised Manuscript. Single *S. cerevisiae* cells budding daughter cells increase mass in linear segments. **a, b**, Mass versus time curves of two *S. cerevisiae* cells segmented (green vertical dashed lines) into phases of linear growth (colored) by a segmented linear regression model (SLM) (violet lines). DIC images show budding cells attached to the cantilever of the picobalance. Shown are raw data. Scale bars, 10 μm . Mass measurements were recorded using the continuous mode. **c, d**, Residuals of fitting the SLM to the raw data of growth curves. **e, f**, Growth rate (GR) of each linear growth segment shown in **a, b**. **g-j**, Growth analysis of 19 cells measured using the continuous mode. **g**, Distribution of the number of linear growth segments measured for all cells. Mean is 5 (orange star) and median (horizontal pink line) is 4 segments per cell with 95% ci=[3.3, 4.7] (dark grey edges of hourglass hexagon). The distribution of the data is shown by the kernel density (light grey area) and the standard deviation (s.d., medium grey vertical box) **h**, Growth rates determined from linear growth segments of all cells. Median is 0.70 pg min^{-1} with 95% ci=[0.6, 0.9]. **i**, Duration (time span) of the linear growth segments of all cells. Median is 13.7 min with 95% ci=[11.4, 16.0]. **j**, Duration of the budding (S/G2/M) phase of all cells. Median is 94 min with 95% ci=[84.1, 104.0].

Reviewer #2 2: Second, mass accumulation of larger cells was previously reported to be faster than for smaller cells (eg Godin et al., 2010 Nat Methods). This is generally assumed to be because cells grow closer to exponentially than to linearly (thereby explaining why a culture of small mutant cells can double in number with close to the same generation time as a culture of larger mutant cells). If, as the authors propose, cells do not grow exponentially it requires that larger cells have a higher linear growth rate than smaller cells, which the authors need to test as a minimum requirement to clarify and corroborate their conclusion.

Authors: We thank the reviewer for pointing out the work of Godin et al., 2010. We kindly point out that Godin et al. do not measure the total (or inertial) mass of a cell but only a fraction of the total mass named buoyant mass. The buoyant mass corresponds to the total mass of a cell minus the mass of the fluid displaced by the cell. Consequently the buoyant mass corresponds to only $\approx 5 - 10\%$ of the total mass of the cell⁸. In Fig. 2C Godin et al. report a start mass of between 3 – 4 pg for *S. cerevisiae* cells. This buoyant mass is in agreement to our measurements of total mass ranging from 76 – 81 pg (**new Fig. 2, new Fig. 3**), which would correspond to a buoyant mass of 3.5 – 4 pg. In fact our measurements also compare well to classical techniques such as dry mass measurements from Mitchison, 1958 (ref⁹). The reported dry mass of ≈ 20 pg would result in 67 pg total mass assuming 70% water content of the cell^{8,9}. Please note that buoyant mass and dry mass are not the same. In addition to above comparison, our picobalance directly measures the total mass of individual cells at ≈ 10 ms time resolution over extended time ranges, which can last from hours to days³. The compelling advantages provided by our picobalance are reported as high total mass and time resolution. To the best of our knowledge there is no reason to assume that the total cell mass (or simply cell mass) has to show the same behaviour as the buoyant mass. This is particularly the case upon considering water transport in and out of the cell. In addition, we kindly note, that direct comparisons between different experiments have to be made with great care since they often do not use the same experimental conditions in terms of strains, media and temperature, all of which being described to affect cell growth and size¹⁰.

The reviewer also suggested to test if larger yeast cells have a higher linear growth rate than smaller cells. To test this we have decided to record and analyse the growth behaviour of more yeast cells (**new Fig. 2, new Fig. 3, new Supplementary Fig. S10, S11,**). By analysing with high time and total mass resolutions the growth behaviour of 38 single yeast cells, we have identified that single yeast cells grow in multiple segments of highly linear growth. Encouraged by the reviewer we have deeply analysed the data and we have found a small-medium positive correlation (Pearson correlation coefficient $r \approx 0.3$) between the growth rate of each growth segment and the cell mass at the segment (**new Supplementary Fig. S10c-e**). Moreover we have identified an even stronger negative correlation, although medium (Pearson correlation coefficient $r = 0.37$), between the duration of the linear segments and the cell mass at the start of the segments (**new Supplementary Fig. S10i**). We have also performed such analysis at the population level (binned and averaged), which shows a much stronger positive correlation between the growth rates with cell mass during the S/G2/M phase (**new Supplementary Fig. S11a-d**) consistent with Godin et al., 2010. We have incorporated this information into the revised Manuscript that now reads “Next, we investigated whether the growth rate of single yeast cells scales with cell size¹¹ and duration of the cell cycle phase^{12–14} such as previously reported. Fitting the SLM model to our raw growth data approximated the growth rate and duration of every linear growth segment (**Fig. 4, Supplementary Fig. S7–S9**). At the single cell level, we found small correlations (Pearson correlation coefficient, $0.1 < r < 0.2$) of the growth rate of the linear segments with the cell mass after the G1/S phase transition

(**Supplementary Fig. S10a**) or with the cell mass at cytokinesis (**Supplementary Fig. S10b**). We found a small-medium correlation ($r \approx 0.3$) of the growth rate of each linear segment with the cell mass at the start, average, or end time points of the segment (**Supplementary Fig. S10c-e**). We found a negligible correlation ($r = 0.08$) between the growth rate of linear segments with the duration of the S/G2/M phase and a small correlation ($r = 0.14$) and negative trend between the growth rate with the duration of each linear growth segment (**Supplementary Fig. S10f,g**). In addition, a negligible correlation was found between the duration of the S/G2/M phase and the cell mass at G1/S transition ($r = 0.08$) (**Supplementary Fig. S10h**). However, the strongest correlation, although being a medium correlation ($r = 0.37$), was a negative trend of the duration of the linear growth segments with the cell mass at the onset of the segment (**Supplementary Fig. S10i**). To summarize, the single cell growth rate increases minimally with the cell mass at the onset of the budding process and weakly depends on the cell mass at any point during the growth. Interestingly, however, the time a single cell spends in a specific linear growth segment depends inversely on its mass.

We also wanted to analyse the growth rate of yeast cells at the population level similar to how it is routinely done upon analysing bulk measurements of cells^{11,15,16}. We hence plotted the instantaneous growth rate *versus* cell mass of single yeast cells at different time points during the S/G2/M phase, and calculated their average growth rate, which showed a medium correlation to the data ($r = 0.4$) and strong correlation ($r = 0.9$) to the average of all single cells. (**Supplementary Fig. S11a,b**). Further binning the cell mass at the G1/S transition in six groups and averaging the growth rate during S/G2/M showed a strong correlation ($r = 0.9$) between cell mass and growth rate (**Supplementary Fig. S11c**). Thus, at the population level our averaged single cell growth data is consistent with literature¹¹ despite the clearly observed segmented linear growth at the single cell level. Further, no correlation between the yeast mass at the G1/S transition and the duration of the S/G2/M phase was observed at the population level (**Supplementary Fig. S11d**). To summarize, upon performing population level analyses, we find in agreement with literature¹¹ that on average cells with a higher mass (size) grow faster. However, at the single cell level we did not find strong correlation ($r > 0.5$) between cell mass and growth rate at G1/S transition. Overall, this finding suggests a more complex growth regulation at the single cell level, which probably would require experiments to be conducted at higher mass and time resolution and using more accurate cell cycle markers.”

New Supplementary Fig. S10 of the revised Manuscript. Growth rate analysis with segmented linear model of mass measurements recorded with single yeast cells during the S/G2/M phase. The data originates from 19 *S. cerevisiae* cells recorded using the high-resolution continuous mode. Each colored circle corresponds to a different single yeast cell. The growth rates (GR) correspond to the slopes of linear segments, which have been identified using a segmented linear regression. Linear fit (red line), 95% prediction interval (dashed lines, magenta) show the general trend. The Pearson correlation (r) and p-Value (p) is shown above the panels. **a**, GR (colored circles) as a function of the mass after Start at the G1/S transition for the S/G2/M (budding) phase. **b**, GR (colored circles) as a function of the mass at cytokinesis (cell division). **c**, GR (colored circles) as a function of the mass at the onset of each linear segment. **d**, GR (colored circles) as a function of the average mass for each linear segment. **e**, GR (colored circles) as a function of the mass at the end of each linear segment. **f**, GR (colored circles) as a function of the total duration of the S/G2/M phase. **g**, GR (colored circles) as a function of the duration of each linear segment. **h**, S/G2/M phase duration (colored circles) as a function of the cell mass at the onset of the S/G2/M phase (G1/S transition) for all 19 measured yeast cells. **i**, Segment duration (colored circles) as a function of the cell mass at the onset of each linear segment.

New Supplementary Fig. S11 of the revised Manuscript. Population analysis shows that on average cells with higher mass at the G1/S transition grow at a higher rate during S/G2/M phase. **a**, Growth rates (GR as dm/dt) of single yeast cells ($n = 19$) plotted against their mass m during their S/G2/M phase (individual mass curves in **Supplementary Fig. S7**). The data was acquired using the continuous mode. Grey circles indicate the data points ($r = 0.4$, $p\text{-Value} = 2.2 \cdot 10^{-05}$), red circles the average and the red line is a fit through the averaged data ($r = 0.9$, $p\text{-Value} = 0 \cdot 10^0$). The slope was found to be 0.0049 min^{-1} . As can be seen, higher masses correlate with higher mass accumulation rates. **b**, As **a** with data ($n = 19$) being acquired using the sweep mode (individual mass curves are shown in **Supplementary Fig. S9**). The slope was found to be 0.0054 min^{-1} . **c**, Mass growth rate of budding yeast in dependence on the start mass as measured at G1/S transition. The average mass growth rate during S/G2/M was plotted for each cell in a box-plot against the start mass of the respective cell. For the plot, the start masses are binned into five bins of equal size. The center bar of the box plot indicates the median, which was used to fit the red trend line. The extend of the box indicates the upper and lower quartile, in which at least 50% of the data lies. The whiskers extend to show the rest of the distribution, as indicated by the 1.5-fold Inter Quartile Range. Based on this categorization, the diamonds are considered outliers. The total mass growth rates correlate with the start mass bins with a correlation coefficient of $r = 0.9$. **d**, S/G2/M phase duration of budding yeast in dependence on the start mass measured at G1/S transition. The S/G2/M phase duration was plotted for each cell in a box-plot against the start mass of the cell. Mass after Start at the G1/S transition binned the same way as in **c**. The duration of the S/G2/M phase does not correlate with the start mass.

Reviewer #2 3: Third, as the authors state when measuring at high resolution after "60 min the budding process stopped (Supplementary Fig. 3 and Movie 4), indicating that the increased laser power impaired cell growth over an extended period of time". This does call in to question whether the growth in the proceeding 60 minutes is likely to be

representative of a truly wild-type growth patterns. The authors need to address this potential issue with their methodology in some way (less laser power?).

Authors: Thank you for your comment. As quoted by the reviewer, the cell mass data shown in the previously submitted **Supplementary Fig. S3** and **Movie S4**, which now corresponds to **Supplementary Fig. S15** and **Movie S6**, has been acquired at ≈ 3 times higher blue laser power ($\approx 28 \mu\text{W}$) compared to all other mass measurements in the manuscript, which have been acquired at $\approx 8 \mu\text{W}$. One aim of the former **Supplementary Fig. S3** was to show that higher mass resolution can be achieved ($\leq 1 \text{ pg}$ at 10 ms time resolution) with the picobalance by increasing the laser power. Our observation that too high blue laser power can harm the yeast cell viability is similar to the observation that phototoxicity can harm the cell viability at too high laser (or light) power in fluorescent microscopy¹⁷. However, monitoring the budding yeast cell at higher laser power (former **Supplementary Fig. S3**) and thus at higher mass resolution allowed to investigate whether the mass increase during the stepwise linear growth correlates to the growth of the bud while the mother cell shows no growth (mass increase). We apologize for the confusion created and have revised our manuscript to avoid confusion by moving the section to **Supplementary Note 3**. Now all measurements shown in the revised main manuscript are compatible with non-invasive conditions (e.g., mass measurements made at $\approx 8 \mu\text{W}$ blue laser power). Characterized under these non-invasive conditions the yeast cells remain fully viable as demonstrated by their ability to generate several generations of daughter cells (**new Fig. 2**, **new Fig. 3**, **new Supplementary Fig. S2, S3, S5**, **Movie S1–S3**). Moreover, the cell viability during the mass measurements is supported by both, our continuous high-resolution mass measurements (**new Fig. 2**, **new Fig 4**, **new Supplementary Fig. S7**) and our discrete mass measurements using the sweep mode (**new Fig. 3**, **new Supplementary Fig. S6, S8, S9**, **Movie S4–S5**). The sweep mode records the mass only throughout intervals of 30 and 50 seconds and thus exposes the cells to much less (blue) light power during the cell cycle because the blue and infrared lasers of the picobalance are switched off in between each mass measurement. The **new Supplementary Fig. S4** shows that the S/G2/M phase durations are indeed the same for the continuous mass measurement mode and the sweep mass measurement mode and are in agreement with literature^{9,13,16,18–21}. Please kindly note that such direct comparison of our data with literature is difficult, since these other studies used different yeast strains, media, and/or biophysical methods to describe cell growth. However, such comparison shows that our data match very well the expected time ranges of the cell cycle phases. We thus believe our presented results indicate that cell growth is not impaired by our measurements and that our measurements are representative of wild-type growth patterns of single *S. cerevisiae* cells. We have clarified this point in our revised manuscript at much better detail and also outlined the rationale of why we performed the high resolution / high laser power experiment (see **revised Discussion**, first section, and **Supplementary Note 3**).

Point-by-Point Response to the Reviewer's Comments

Reviewer #3

Reviewer #3: Two years ago Martinez-Martin and co-workers introduced the inertial picobalance (8) to weigh and detect mass fluctuations on adherent cells. That paper was a breakthrough in nanomechanical sensing. The inertial picobalance enables to follow in real time the mass changes of a living cell. In that paper, the authors took great care in designing a very sensitive device compatible with a single cell biology.

The current manuscript is a follow up. The mass sensitivity of the inertial picobalance has been improved by reducing the total mass of the cantilever. This was achieved by reducing the cantilever size. The current inertial picobalance is more reliable, robust and sensitive than the first prototype. This evolution might be necessary for commercial or pre-commercial applications (refs. 31-33). However, it seems that all the necessary physics and/or technology involved in reducing the cantilever size has been borrowed from other fields (AFM).

The technology associated with the present contribution is outstanding. Moreover, the authors findings are explained with clarity in the manuscript. Nonetheless, I find it hard to determine how this paper will influence thinking in the field. This manuscript relies heavily of their previous results. The fact that the mass sensitivity has been improved to the picogram range is an achievement. However, it is far from surprising. Nanomechanical resonators fabricated by top-down approaches have achieved mass sensitivities in the zeptogram range in vacuum (Yang, Y. T. et al. Nano Lett. 6, 583–586 (2006)). Closer to the manuscript context, ref. 28 claims a mass resolution in the sub-femtogram range. In my view, the novelty of the manuscript rests on the relevance of the observed stepwise growth of a budding yeast cell. The authors should provide more details about this point.

Authors: Thank you for your encouraging comments. As mentioned by the reviewer, we kindly acknowledge that nanomechanical resonators have been developed to achieve much higher mass resolution approaching the zeptogram range in vacuum²² or femtogram range in air²³. To measure the mass of living cells these resonators have microchannels filled with solution. However, while such microchannels resonating in vacuum allow to measure the buoyant mass of single suspended cells they do not allow to measure their total mass directly²⁴. In addition, the techniques do not allow to monitor the mass of the same single living cell at high time (ms) resolution over extended periods of times (hours to days). We also kindly note that our technologically improved picobalance is not only based on the manufacturing of smaller cantilevers, which were needed to achieve a higher mass resolution, but also on using ultra-low laser power to photoactuate the cantilever at conditions that do not perturb the viability of the photosensitive yeast cells. We further developed the data processing procedures, which allow for a ≤ 0.5 pg mass accuracy at a time resolution of 100 s. Overall we increased the mass resolution by a factor of ≈ 11 , while reducing the laser power six times, which allows yeast cells to grow unperturbed on the cantilever, while they are continuously weighted and morphologically observed. With

these technological advancements our current total mass resolution is ≈ 20 times higher than the total mass resolution achieved so far by pedestal mass sensors⁴ and approaches the femtogram range.

Moreover, being part of our revision, we redesigned the cantilever holder, the optical path and the cell culture chamber of the picobalance to allow the detection of the relatively low fluorescent signal of proteins reporting the cycle phases of single yeast cells. Taken together, we thus believe that the technological advancements presented in our manuscript considerably advance the field, as they introduce new ways to continuously determine the total mass of single living yeast cells at much improved mass resolution in real-time and at cell culture conditions while tracking their cell cycle using fluorescence microscopy. Our approach is technologically very different compared to other approaches, which either measure the buoyant mass of a living cell, which represents a fraction of the total mass of the cell, or the mass of inert (dried cells) samples in vacuum^{11,24}.

Encouraged by the reviewer's suggestion and by the substantial amount of additional experimental data recorded to revise our Manuscript, we have now outlined at better detail the technological differences and advancements of the picobalance, which allow us to monitor the total mass of individual yeast cells progressing through the cell cycle. The new experimental data and analysis reveal novel insight on the segmented linear growth behaviour of single yeast cells processing through the S/G2/M phase. We also highlighted the relevance and implications of a segmented linear growth model for the current perception of the field regarding cell growth during the cell cycle (see **revised Manuscript, Abstract, Introduction, Results, Discussion and Methods**). Moreover, in our **revised Discussion** we discuss the importance of single cell measurements to unveil their individual growth behaviours. Below please find our point-by-point response to the specific referee comments, which guided us to revise our Manuscript.

Reviewer #3: Comments. 1 The field of nanomechanical sensors has 20 years of history. Cantilever-based sensors have been applied to measure biological interactions and mass changes (DNA, proteins, cells). There are some good reviews (for example, J. Tamayo et al. Chem. Soc. Rev. (2014)) and original contributions. Some of those references could strengthen the context of the manuscript.

Authors: We thank the reviewer for suggesting an excellent review. We have included the review and other citations to reflect the state of the field into our revised Manuscript (see **revised Introduction**) to better introduce the field of nanomechanical sensors and their application to Biology.

Reviewer #3: 2 Unprecedented accuracy is repeated several times. It appears twice in the abstract.

Authors: Thank you. We apologize for the redundancy. We have revised our Manuscript to reduce redundancy to a minimum.

Reviewer #3: 3 Photothermal excitation can be accomplished with a range of light visible wavelengths (blue, red...). If there is a technical/scientific reason for using blue light, this should be mentioned.

Authors: As the reviewer mentions, photothermal excitation can be performed using light having different wavelengths. We have chosen blue (centre wavelength of 405 nm) light in our device to excite the cantilever for the following two reasons:

1. Blue light photothermally actuates silicon cantilevers very efficiently without the need of metallic coatings. The absorption coefficient of the cantilever material silicon for blue light (405 nm) is one order of magnitude larger than for red light (680 nm) and two orders of magnitude larger than for infrared light (≈ 800 nm)²⁵.
2. Blue light is far away from the emission spectrum of common fluorophores and hence it is easy to filter out.

We have now included this information in the **revised Manuscript** (section **revised Methods**), which reads *“The blue laser brings along the advantage to efficiently photoactuate silicon (or silicon nitride) cantilevers even in the absence of metallic coatings, while the optical absorption coefficient of silicon decreases for longer wavelengths²⁵. Because the wavelength of the blue laser is far away from the emission spectrum of common fluorophores, it enhances the compatibility of the picobalance with fluorescent microscopy.”*

Reviewer #3: 4 In page 12 is stated that ‘...this stepwise linear growth...’. The terms ‘stepwise’ and ‘linear’ are at odds. Stepwise means it proceeds in steps (finite jumps). Linear means proportional to a parameter (here time).

Authors: We thank the reviewer for pointing out this unclear description. We have modified the wording and now use a segmented linear regression model (SLM), which is mathematically the standard terminology to describe a segmented function whose behaviour is linear in every segment. The revised sentence reads in the **revised Manuscript** (section **Discussion**): *“By directly monitoring the total cell mass, morphology and cell cycle phase, we observe that single *S. cerevisiae* cells progressing through the S/G2/M phase - in which they bud - grow in multiple linear segments (**Fig. 4, 5, Supplementary Fig. S7–S9**). Each linear growth segment shows a slope (growth rate) higher or lower than that of the previous segment and differ in timespan. For instance, some linear growth segments show a slope approaching to zero and hence no growth, while other segments show steep mass increases. On average, single yeast cells progressing through the S/G2/M phase show five distinct linear growth segments.”*

Reviewer #3: 5 Similarly, the meaning of void slope is obscure. Do the authors mean a negative slope (Fig. 3)? Flat slope?

Authors: We thank the reviewer for helping us to improve the description of the mass behaviour. By void slope we meant slope equal to zero. As we have already implemented in the previous question, we have now changed “void slope” by “slope approaching to zero and hence no growth”. The former **Fig. 3** has been replaced by **new Fig. 4** and extended by new measurements shown in **new Supplementary Fig. S8**. The new data nicely shows the distribution of the growth rates (slopes derived from fitting linear segments of constant growth rate) measured from individual cells (**new Fig. 4h, new Supplementary Fig. S8h**). The former **Fig. 3** is now **new Supplementary Fig. 3**.

Reviewer #3: 6 Figure 3, could the authors the place an arrow at the times that there is a step in the growth ?.

Authors: Thank you. The former **Fig. 3** has been revised and replaced by the **new Fig. 4**. The regions indicating the linear growth segments of the cell are shown in different colours. We also show additional data of the single cells in **new Supplementary Fig. S7–S9** where the linear growth segments have been separated by vertical green dashed lines and fitted by a segmented linear model (violet lines). We believe these representations are more intuitive than arrows. The former **Fig. 3** is now **new Supplementary Fig. 3** with the linear growth segments of constant growth rate being separated by vertical green dashed lines as well.

New Fig. 4 of the revised Manuscript. Single *S. cerevisiae* cells budding daughter cells increase mass in linear segments. a, b, Mass versus time curves of two *S. cerevisiae* cells segmented (green vertical dashed lines) into phases of linear growth (colored) by a segmented linear regression model (SLM) (violet lines). DIC images show budding cells attached to the cantilever of the picobalance. Shown are raw data. Scale bars, 10 μm . Mass measurements were

recorded using the continuous mode. **c, d**, Residuals of fitting the SLM to the raw data of growth curves. **e, f**, Growth rate (GR) of each linear growth segment shown in **a, b**. **g-j**, Growth analysis of 19 cells measured using the continuous mode. **g**, Distribution of the number of linear growth segments measured for all cells. Mean is 5 (orange star) and median (horizontal pink line) is 4 segments per cell with 95% ci=[3.3, 4.7] (dark grey edges of hourglass hexagon). The distribution of the data is shown by the kernel density (light grey area) and the standard deviation (s.d., medium grey vertical box) **h**, Growth rates determined from linear growth segments of all cells. Median is 0.70 pg min⁻¹ with 95% ci=[0.6, 0.9]. **i**, Duration (time span) of the linear growth segments of all cells. Median is 13.7 min with 95% ci=[11.4, 16.0]. **j**, Duration of the budding (S/G2/M) phase of all cells. Median is 94 min with 95% ci=[84.1, 104.0].

Reviewer #3: 7a How many yeast cells show a stepwise mass growth?

Authors: We have now included additional data comprising a total of 38 different yeast cells. 19 out of 19 yeast cells monitored in the high-resolution continuous mode show multiple segments of linear growth (**new Fig. 4g, Supplementary Fig. S7**) and 13 out of 19 yeast cells monitored in the sweep mode show multiple segments of linear growth (**new Supplementary Fig. S8g, S9**). Please note that the data recorded in the sweep mode does not have the high resolution as obtained in the continuous mode and consequently can reveal fewer segments of linear growth.

Reviewer #3: 7b Is Fig. 3 representative of the growth cycle of budding yeast cells?

Authors: Thank you, we have replaced the former **Fig. 3** by new experimental data (**new Fig. 2a,b, 3a,b, Supplementary Fig. S2, S3, S5, Movie S1–S5**), which all show representative growth cycles of budding yeast cells and led to a much improved analysis and model of yeast cell growth (**new Fig. 4, 5**). As supported by the new additional data, the mass behaviour depicted in revised **new Fig. 4** is representative of the growth cycle of budding yeast cells, as it shows that single yeast cells increase total mass in linear segments over time. However, the number and time span of linear growth segments as well as their slopes can vary among individual yeast cells. The former **Fig. 3** is now shown as **Supplementary Fig. S3**.

New Fig. 2 of the revised Manuscript. Mass and morphology of single *S. cerevisiae* cells budding daughter cells.

a, b, While continuously measuring the total mass of a single yeast cell, differential interference contrast (DIC) images (taken every 5 min) show the budding of the cell attached to the cantilever of the picobalance (**Supplementary Movie S1**). White arrows indicate budding daughter cells. The raw data (black curve) shows the total mass of a growing yeast cell acquired every 10 ms using the continuous mode. The red curve shows the average raw data (100 s moving window, **Methods**). Cyan bars indicate where cells are in the S/G2/M phase when bud growth is observed. Yeast cells were attached to ConA-coated microcantilevers and the measurements were recorded in yeast culture medium at 30°C (**Methods**). **c,** Growth curves ($n = 19$) of single yeast cells in the S/G2/M phase (bud growth) as measured by the picobalance using the continuous mode (moving average of 100 s). The overall growth rates range between 0.2 pg min^{-1} and 1.6 pg min^{-1} , with an average of $0.7 \pm 0.3 \text{ pg min}^{-1}$ (mean \pm s.d.). The duration of the S/G2/M phase ranges from 50 min to 140 min, with an average of $93 \pm 21 \text{ min}$ (**Supplementary Fig. S2**). DIC images in **a** and **b** were contrast enhanced using a custom flat field correction (**Methods**). Scale bars, 10 μm .

New Fig. 3 of the revised Manuscript. Mass and cell cycle measurements of single *S. cerevisiae* cells budding daughter cells. a, b, Single yeast cells expressing the fluorescent cell cycle markers (Myo1-mKate(3x) and Whi5-mKOkappa(1x)), were imaged using DIC and fluorescence microscopy every 2 min (upper panels). A phase and amplitude curve of the microcantilever were recorded over intervals ≈ 50 s to measure the cell mass using the sweep mode (**Supplementary Movie S4**). Between consecutive mass measurements, the infrared and blue lasers of the picobalance were switched off for ≈ 20 s to reduce bleaching of the fluorophores and to reduce potential perturbation of yeast growth. Cell mass values as derived from sets of single amplitude curves are shown as grey dots. Average raw data (350 s moving window, red line) shows the trend. Cyan bars on the time axis denote the S/G2/M phase of the yeast cell cycle and magenta bars denote the G1 phase. The star (*) in **b** denotes the (partial) detachment of the daughter cell after cytokinesis, which drops the total mass. **c**, Growth curves ($n = 19$) of single yeast cells progressing through the S/G2/M phase (bud growth) as measured by the picobalance using the sweep mode. Scale bars, 10 μm .

Reviewer #3: 7c I might be wrong, but Figure 4 does not seem to correlate with the pattern shown in Fig. 3 (time: 15 to 45 minutes).

Authors: Former **Fig. 3 (now Supplementary Fig. S3)** and former **Fig. 4 (now Supplementary Fig. S13)** showed two different yeast cells from two different experiments, which is visible from the associated optical images. Therefore, the patterns do not need to correlate. Former **Fig. 4 (now Supplementary Fig. S13)** is, however, a subset (zoom) of **new Supplementary Fig. S15** and should correlate, which it does.

Reviewer #3: 3 Photothermal excitation can be accomplished with a range of light visible wavelengths (blue, red...). If there is a technical/scientific reason for using blue light, this should be mentioned.

Authors: This question seems to be duplicated and has been answered above.

Point-by-Point Response to the Reviewer's Comments

Reviewer #4

Reviewer #4: In this manuscript the authors describe the development of a molecular 'balance', capable of continuously measuring the mass of single yeast cells at pg resolution. They use the information they gathered from monitoring in real time dividing cells to arrive at the conclusion that yeast cells increase in mass linearly, not exponentially as it is often assumed. If true, these results have important implications for current models of size control and how cell growth is coupled to division, because a linear increase in mass (documented with the most accurate instrument possible) questions the need/existence for critical size controls during cell division. Overall, the authors appear to have significantly 'moved the needle' in the field regarding the instrumentation they developed to measure the mass of small cells. This is the main reason for being excited about the paper and its potential impact. However, I have several concerns:

Authors: Thank you for your encouraging comments. Below please find our point-by-point response to each of your concerns.

Reviewer #4: 1. It was not clear that the authors measured the increase in mass in the G1 phase of the cell cycle. It appears that for single cells their measurements begun at the time of bud emergence (before that and during attachment to the cantilever the cells did not grow in their setup). If true, their measurements essentially exclude the G1 phase. This is crucial, because much of the interest in the linear vs. exponential model of growth is in G1.

Authors: Thank you. We have revised the sections to more precisely describe which phase of the cell cycle was measured when. Our focus was indeed on the S/G2/M phase, which we could initially identify in the absence of cell cycle markers from bright field images recorded during the mass measurements. We strongly believe that our findings that the cells in S/G2/M phase grow in multiple linear segments, which can differ in their number, time span and slope, is of great importance for the scientific field since it directly affects current cell cycle models^{11,13,26-30}. In principle, we could also measure the G1 phase as we typically started our mass measurements with newly born yeast cells. However, it is true that some of these measurements rather characterized the late G1 phase or close to transit towards the G1 and S/G2/M phase. The comment of the reviewer has motivated us to further develop the picobalance so that we can now measure the mass measurements and fluorescence of single yeast cells simultaneously. To determine the yeast cell cycle phases more precisely, we used a new strain bearing two cell cycle markers, the nuclear re-localisation marker Whi5 served as a proxy for the G1 phase while the bud neck marker Myo1 served as a proxy for S/G2/M phase⁷. To be able to measure the G1 phase we had to face several issues. One issue was that the time needed to optically find/select a cell in the right state and to mechanically attach the cell to the

cantilever proved to be too long to capture the full G1 phase. Another issue was that the sweep mode does not provide high enough mass resolution, while the continuous mode is not yet working with fluorescence. The **new Fig. 3a-b** shows a yeast cell growing over several phases including the G1 phase. The new figure, however, also highlights the partial detachment of the budding cell after cell division, which impairs the mass measurements with our cantilever during the G1 phase. In contrast, a full detachment of the daughter cell would allow to measure a full second G1 phase (**new Supplementary Fig. S5, Movie S5**). In summary, the quality and statistics of the data we could so far gain for the G1 phase is insufficient for a proper statement. Nevertheless, we now include data of the G1 phase in **new Supplementary Fig. S6** to at least show a trend. A representative growth curve including G1 measured with continuous mode but without fluorescence is shown in **new Supplementary Fig. S2, S3**. However, in the future a substantial revision of the experimental setup and pick up protocol (**Methods, Supplementary Fig. S12**) might enable us to address the exciting question of growth in the G1 phase.

New Fig. 3 of the revised Manuscript. Mass and cell cycle measurements of single *S. cerevisiae* cells budding daughter cells. a, b, Single yeast cells expressing the fluorescent cell cycle markers (Myo1-mKate(3x) and Whi5-mKOkappa(1x)), were imaged using DIC and fluorescence microscopy every 2 min (upper panels). A phase and amplitude curve of the microcantilever were recorded over intervals ≈ 50 s to measure the cell mass using the sweep mode (**Supplementary Movie S4**). Between consecutive mass measurements, the infrared and blue lasers of the picobalance were switched off for ≈ 20 s to reduce bleaching of the fluorophores and to reduce potential perturbation of yeast growth. Cell mass values as derived from sets of single amplitude curves are shown as grey dots. Average raw data (350 s moving window, red line) shows the trend. Cyan bars on the time axis denote the S/G2/M phase of the yeast cell cycle and magenta bars denote the G1 phase. The star (*) in **b** denotes the (partial) detachment of the daughter cell after cytokinesis, which drops the total mass. **c,** Growth curves ($n = 19$) of single yeast cells progressing through the S/G2/M phase (bud growth) as measured by the picobalance using the sweep mode. Scale bars, 10 μm .

New Supplementary Fig. S6 of the revised Manuscript. Mass curves recorded of single *S. cerevisiae* cells in the G1 phase. **a**, Mass curves for single yeast cells progressing through the G1 phase acquired using the consecutive sweep mode. The time 0 min corresponds to the start of the G1 phase (Whi5 fluorescent signal in nucleus). The cell cycle phase was determined by fluorescence. **b**, Mass curves in **a** normalized to their respective end mass. The time 0 min corresponds to the beginning of S phase.

Reviewer #4: 2. Their value of 92 pg/cell at the beginning of the experiment appears too large. Certainly much larger than the value of ~20 pg in the paper they cite by Mitchison's measurements of dry mass. I guess adding the missing liquid will bring that value closer to ~60 pg, but still their 92 pg value appears too high.

Authors: Although a mass of 92 pg may appear too high at first glance, the value is reasonable for a yeast having a diameter of $\approx 5.4 \mu\text{m}$ (**new Supplementary Fig. S3**), which is compatible with literature values³¹. Please note that for yeast cells having a spherical shape, small changes of the radius lead to large changes in mass, which scales with the cubic of the radius (**Supplementary Note 1, new Supplementary Fig. S13**). A comparison with the literature reveals that yeast cells measured in similar media (YNB + 2% glucose) as used in our studies have volumes of $(64.6 \pm 13.3) \mu\text{m}^3$ (ref²⁷). Assuming a density of 1.1 g cm^{-3} for *S. cerevisiae*^{8,32} such diameter results in an estimated total mass of 90.7 pg. More recent studies report the diameter of yeast cells in the G1 to S/G2/M transition to range between 5 and $6 \mu\text{m}$ ³¹, which suggests the total mass of a *S. cerevisiae* cell to range between 71.0 and 124.0 pg. However, a direct comparison of the cell masses estimated from different experiments is difficult, because it is expected that the volume and thus the total mass of yeast cells depend on the culture media, strain, temperature, and method used to estimate the cell volume and/or mass. Nevertheless, the cell (total) mass values of our measurements ($\approx 40 - 150 \text{ pg}$, **new Fig. 2, new Fig. 3, new Supplementary Fig. S7–S9**) agree well with those estimated from other approaches. Finally, the reported dry mass from Mitchison's measurements of $\approx 20 \text{ pg}$ would result in $\approx 67 \text{ pg}$ total mass assuming 70% water content of the cell^{8,9}. However, 60 – 80% water content have been reported which would lead to a range between 50 – 100 pg total mass based on the $\approx 20 \text{ pg}$ dry mass^{8,9}.

Reviewer #4: 3. The authors need to check more strains and media to further validate their results. For example, A haploid vs. diploid strain in rich vs. poor media will provide important reference points.

Authors: Our picobalance is a single cell technique, which makes it very challenging to conduct a high number of measurements such as needed to widely screen conditions. For each yeast cell measurement we have to produce (micro-machining using a focused ion

beam (FIB)) several cantilevers among which we have to select and calibrate the cantilevers that can be efficiently photothermally actuated with just a few μW . After this, we have to pick up a yeast cell with the cantilever and then conduct mass measurements. The latter takes several hours to characterize the growth of one yeast cell (and its progeny). Thus to test the growth of different yeast strains in different media would take at least 6 months of experiments, which are then followed by data analysis and iterations of follow up measurements. Hence, although the suggested comparison between the behaviour of haploid *versus* diploid strains as well as the exposure of yeast cells to different media would be very interesting, we think that widely screening these various conditions is beyond the scope of our manuscript.

Nevertheless, being encouraged by the reviewer's comment, we have recorded and included a substantial amount of additional data of growing yeast cells. While one set of the new data has been acquired, using the same yeast strains and media as initially presented (**new Fig. 2, new Supplementary Fig. S7**), the other data set has been recorded using fluorescently Myo1 and Whi5 tagged yeast cells to monitor cell mass and cycle of the growing yeast cell (**new Fig. 3, new Supplementary Fig. S9**).

Lastly, before screening the relevance of different media and yeast strains more work on the technological side is required to find a suitable way to monitor all cell cycle phases properly. This should include the full first G1 phase of the mother cell after being born (replicative age 0), followed by the S/G2/M phase during which a new daughter cell is being born. After cell division, both cells enter the G1 phase. It would be of great interest to measure the mass of mother cells with higher replicative age (≥ 1) since it has been reported that cell age has an effect on the cell cycle duration³³. Currently, such recording is not possible because it is unknown how a daughter cell partially detaching from the mother cell contributes to the total mass measured (**Supplementary Fig. S5, Movie S5 vs Supplementary Fig. S2, S3, Movie S2–S3**). Additionally, both cells can progress through the G1 phase in different times. Therefore, before executing such experiments, first the technological setup and protocol has to be further developed.

Reviewer #4: 4. The authors do not mention some key references:

- For the significance of linear vs. exponential growth in terms of cell size control, perhaps one of the most lucid papers is the now classic Conlon and Raff paper (PMID: 12733998). A plethora of newer papers, especially as they relate to the 'adder' phenomenon should also be discussed.
- For linear growth (measured in dry mass) they cite the 1958 Mitchison paper, perhaps the 1964 paper from Scopes and Williamson (PMID14195443) should also be cited.
- Linear growth (in size), was also observed by the Heinemann group with a microfluidic device during the entire lifespan of yeast cells (PMID22421136).
- On the other hand, the Manalis group showed that growth (in size), especially in G1, is exponential (PMID22272256).

Authors: Thank you, the references have been included into the revised Manuscript.

References

1. Albrecht, T. R., Grutter, P., Horne, D. & Rugar, D. Frequency-Modulation Detection Using High-Q Cantilevers for Enhanced Force Microscope Sensitivity. *J. Appl. Phys.* **69**, 668–673 (1991).
2. Umeda, N., Ishizaki, S. & Uwai, H. Scanning attractive force microscope using photothermal vibration. *J. Vac. Sci. Technol. B Microelectron. Nanom. Struct.* **9**, 1318 (1991).
3. Martinez-Martin, D. *et al.* Inertial picobalance reveals fast mass fluctuations in mammalian cells. *Nature* **550**, 500–505 (2017).
4. Park, K. *et al.* Measurement of adherent cell mass and growth. *Proc. Natl. Acad. Sci. U. S. A.* **107**, 20691–20696 (2010).
5. Alsteens, D. *et al.* Nanomechanical mapping of first binding steps of a virus to animal cells. *Nat. Nanotechnol.* **12**, 177–183 (2017).
6. Martinez-Martin, D., Alsteens, D., Müller, D. J., Martin, S. & Fläschner, G. Top-cover for a controlled environmental system, top-cover-set and controlled environmental system compatible with probe based techniques and procedure to control the environment for a sample. WO/2017/01 (2018).
7. Talia, S. Di, Skotheim, J. M., Bean, J. M., Siggia, E. D. & Cross, F. R. The effects of molecular noise and size control on variability in the budding yeast cell cycle. *Nature* **448**, 947–951 (2007).
8. Neurohr, G. E. & Amon, A. Relevance and Regulation of Cell Density. *Trends Cell Biol.* **30**, 213–225 (2020).
9. Mitchison, J. M. The Growth of Single Cells II. *Saccharomyces Cerevisiae*. *Exp. Cell Res.* **15**, 214–221 (1958).
10. Johnston, G. C., Ehrhardt, C. W., Lorincz, A. & Carter, B. L. A. Regulation of cell size in the yeast *Saccharomyces cerevisiae*. *J. Bacteriol.* **137**, 1–5 (1979).
11. Godin, M. *et al.* Using buoyant mass to measure the growth of single cells. *Nat. Methods* **7**, 387–390 (2010).
12. Chandler-Brown, D., Schmoller, K. M., Winetraub, Y. & Skotheim, J. M. The Adder Phenomenon Emerges from Independent Control of Pre- and Post-Start Phases of the Budding Yeast Cell Cycle. *Curr. Biol.* **27**, 2774–2783 (2017).
13. Allard, C. A. H., Decker, F., Weiner, O. D., Toettcher, J. E. & Graziano, B. R. A size-invariant bud-duration timer enables robustness in yeast cell size control. *PLoS One* **13**, 1–15 (2017).
14. Heldt, F. S., Lunstone, R., Tyson, J. J. & Novák, B. Dilution and titration of cell-cycle regulators may control cell size in budding yeast. *PLoS Comput. Biol.* **14**, 1–26 (2018).
15. Mir, M. *et al.* Optical measurement of cycle-dependent cell growth. *Proc. Natl. Acad. Sci. U. S. A.* **108**, 13124–13129 (2011).
16. Ferrezuelo, F. *et al.* The critical size is set at a single-cell level by growth rate to attain homeostasis and adaptation. *Nat. Commun.* **3**, 1–11 (2012).
17. Schmidt, G. W., Cuny, A. P. & Rudolf, F. Preventing photomorbidity in long-term multi-color fluorescence imaging of *saccharomyces cerevisiae* and *S. Pombe*. *G3 Genes, Genomes, Genet.* **10**, 4373–4385 (2020).
18. Jonas, F. *et al.* A Visual Framework for Classifying Determinants of Cell Size. *Cell Rep.* **25**, 3519–3529 (2018).
19. Qu, Y. *et al.* Cell Cycle Inhibitor Whi5 Records Environmental Information to Coordinate

- Growth and Division in Yeast. *Cell Rep.* **29**, 987–994 (2019).
20. Barber, F., Amir, A., Murray, A. W. & Murray, A. W. Cell-size regulation in budding yeast does not depend on linear accumulation of Whi5. *Proc. Natl. Acad. Sci. U. S. A.* **117**, 14243–14250 (2020).
 21. Garmendia-Torres, C., Tassy, O., Matifas, A., Molina, N. & Charvin, G. Multiple inputs ensure yeast cell size homeostasis during cell cycle progression. *Elife* **7**, 1–27 (2018).
 22. Yang, Y. T., Callegari, C., Feng, X. L., Ekinici, K. L. & Roukes, M. L. Zeptogram-scale nanomechanical mass sensing. *Nano Lett.* **6**, 583–586 (2006).
 23. Burg, T. P. *et al.* Weighing of biomolecules, single cells and single nanoparticles in fluid. *Nature* **446**, 1066–1069 (2007).
 24. Popescu, G., Park, K., Mir, M. & Bashir, R. New technologies for measuring single cell mass. *Lab Chip* **14**, 646–652 (2014).
 25. Green, M. A. & Keevers, M. J. Optical Properties of Intrinsic Silicon at 300 K. *Prog. Photovoltaics Res. Appl.* **3**, 189–192 (1995).
 26. Leitao, R. M. & Kellogg, D. R. The duration of mitosis and daughter cell size are modulated by nutrients in budding yeast. *J. Cell Biol.* **216**, 3463–3470 (2017).
 27. Jorgensen, P. *et al.* A dynamic transcriptional network communicates growth potential to ribosome synthesis and critical cell size. *Genes Dev.* **18**, 2491–2505 (2004).
 28. Schmoller, K. M., Turner, J. J., Kõivomägi, M. & Skotheim, J. M. Dilution of the cell cycle inhibitor Whi5 controls budding-yeast cell size. *Nature* **526**, 268–272 (2015).
 29. Hartwell, L. H. & Unger, M. W. Unequal Division in *Saccharomyces Cerevisiae* and its Implications for the Control of Cell Division. *J. Cell Biol.* **75**, 422–435 (1977).
 30. Johnston, G. C., Pringle, J. R. & Hartwell, L. H. Coordination of growth with cell division in the yeast *Saccharomyces cerevisiae*. *Exp. Cell Res.* **105**, 79–98 (1977).
 31. Maria, A. *et al.* Real-time monitoring of the budding index in *Saccharomyces cerevisiae* batch cultivations with in situ microscopy. *Microb. Cell Fact.* **17**, 1–12 (2018).
 32. Baldwin, W. W. & Kubitschek, H. E. Buoyant Density Variation during the Cell-Cycle of *Saccharomyces-Cerevisiae*. *J. Bacteriol.* **158**, 701–704 (1984).
 33. Janssens, G. E. & Veenhoff, L. M. The natural variation in lifespans of single yeast cells is related to variation in cell size, ribosomal protein, and division time. *PLoS One* **11**, 1–18 (2016).

Reviewers' Comments:

Reviewer #2:

Remarks to the Author:

The authors have addressed my concerns and I recommend publication.

Reviewer #3:

Remarks to the Author:

The author have addressed my comments. The manuscript suitable for publication.

Reviewer #4:

Remarks to the Author:

I think the authors did a good job with the revisions, adding a significant amount of data. I support publication of this work because it is important to the field to have such measurements available. However, does the work change our current view of the problem? Minimally so, in my opinion. With the issues in measuring growth patterns in G1 (the revisions notwithstanding), what we are left with is very accurate measurements of the S,G2,M and the strong conclusion that growth is largely linear during those phases. But that was already known (see PMID: 17713537). In any case, this is solid work that will strenghten the field.

Manuscript NCOMMS-19-14824A "High-resolution mass measurements of single budding yeast reveal linear growth segments" Cuny et al.

Point-by-Point Response to the Reviewer's Comments

Reviewer #2

Reviewer #2: The authors have addressed my concerns and I recommend publication.

Authors: Thank you for your encouraging and constructive comments, which have guided us to improve our revised manuscript.

Reviewer #3

Reviewer #3: The author have addressed my comments. The manuscript suitable for publication.

Authors: Thank you for your encouraging and constructive comments, which have guided us to improve our revised manuscript.

Reviewer #4

Reviewer #4: I think the authors did a good job with the revisions, adding a significant amount of data. I support publication of this work because it is important to the field to have such measurements available. However, does the work change our current view of the problem? Minimally so, in my opinion. With the issues in measuring growth patterns in G1 (the revisions notwithstanding), what we are left with is very accurate measurements of the S,G2,M and the strong conclusion that growth is largely linear during those phases. But that was already known (see PMID: 17713537). In any case, this is solid work that will strengthen the field.

Authors: Thank you for your encouraging and constructive comments, which have guided us to improve our revised manuscript.